# NEUTAG: Graph Transformer for Attributed Graphs

## Abstract

Graph Transformers (GT) have demonstrated their superiority in graph classification tasks, but their performance in node classification settings remains below par. They are designed for either homophilic or heterophilic graphs and show poor scalability to million-sized graphs. In this paper, we address these limitations for node classification tasks by designing a model that utilizes a special feature encoding that transforms the input graph separating nodes and features, which enables the flow of information not only from the local neighborhood of a node but also from distant nodes, via their connections through shared feature nodes. We theoretically demonstrate that this design enables each node to exchange information with all other nodes in the graph, effectively mimicking all-node-pair message passing while avoiding $\mathcal{O}(N^2)$ computational complexity. We further analyze the universal approximation bounds of the proposed transformer. Finally, we demonstrate the effectiveness of the proposed method on diverse sets of large-scale graphs, including the homophilic & the heterophilic varieties.

## 1 Introduction

Graph neural networks (GNN) (Hamilton et al., 2017; Veličković et al., 2018; Xu et al., 2019; Abu-El-Haija et al., 2019) are increasingly considered de facto models for solving graph mining tasks such as graph classification, node classification, link prediction, etc. Recent advances in the transformer (Vaswani et al., 2017) family of neural networks, especially in the domain of language (Devlin et al., 2019; Radford et al., 2019), and vision (Dosovitskiy et al., 2020) have propelled their applications in the graph domain as well, specifically for graph classification tasks (Rampášek et al., 2022). Graph Transformers (GT) take node sequences as input, along with their attributes, structural and positional encodings, and apply transformer layers successively to learn contextual node representations (Vaswani et al., 2017). It enables modeling long-range dependencies among nodes, avoids over-smoothing (Liu et al., 2020) problems linked with deeper GNN, and is more expressive due to structural (Dwivedi et al., 2022a) and position encodings (Kreuzer et al., 2021a; Dwivedi et al., 2023a). These advantages in the GT architecture often lead them to outperform other GNN-based methods, especially in molecular and biological graph classification tasks, as shown in GraphGPS (Rampášek et al., 2022).

However, the sizes of graphs considered in graph classification tasks are typically of small scale, that is, $\approx 100$ nodes per graph. In contrast, node classification tasks often involve graphs with millions of nodes, such as snap-patents (Lim et al., 2021). One of the fundamental limitations of utilizing graph transformers in these settings is dense attention, which computes attention among all node pairs, leading to $\mathcal{O}(N^2)$ computation in each layer. This is computationally prohibitive and not practical for applications involving large graphs. Recently, sparse-attention methods (Choromanski et al., 2020; Zaheer et al., 2021) proposed in language models have been utilized in GT (Rampášek et al., 2022) to approximate dense attention. However, these sparse-attention methods don't explicitly leverage the structural properties of graphs, resulting in suboptimal performance compared to dense attention-based graph transformers. Recently, graph-specific sparse transformers (Rampášek et al., 2022; Shirzad et al., 2023; Kong et al., 2023; Chen et al., 2022b; Liu et al., 2023; Zhu et al., 2024) have been proposed to incorporate graph topology which learn virtual tokens via global, anchor nodes or clustering. However, none of these methods explicitly leverage the feature-dimension as carrier of long-range communications.

## 1.1 Existing Works and their limitations

A plethora of graph transformers have been proposed to target many aspects of representation learning. A recent survey (Müller et al., 2024) characterizes these key innovations into four primary dimensions: 1) the design of positional and structural encodings (Chen et al., 2022a; Dwivedi et al., 2022a; Bouritsas et al., 2021; Kreuzer et al., 2021b; Lim et al., 2023; Dwivedi & Bresson, 2021; Wang et al., 2022), 2) handling of geometric vs non-geometric features (Fuchs et al., 2020; 2021; Shi et al., 2023; Luo et al., 2023), 3) graph tokenization (Kim et al., 2022; Hussain et al., 2022; Chen et al., 2022b), and 4) propagation mechanisms (Rampášek et al., 2022; Shirzad et al., 2023; Kong et al., 2023; Chen et al., 2022b; Dwivedi et al., 2023b; Ma et al., 2023) (Ma et al.; Liu et al., 2023; Kuang et al., 2021; Zhu et al., 2024; Liu et al., 2023). We have additionally identified a fifth dimension focusing on replacing standard self-attention formulae with equivalent, scalable formulations using linear, polynomial and diffusion processes based kernels (Wu et al., 2023a;b; Deng et al., 2024). These improve scalability but at the cost of expressivity Deng et al. (2024) and still require graph partitioning on million-sized graphs, breaking all-pair connectivity.

This paper focuses primarily on the fourth category, which aims to approximate all pair attention connectivity. These methods typically adopt a common modular architecture. Each layer is composed of a message-passing neural network (MPNN) (Hamilton et al., 2017; Veličković et al., 2018; Xu et al., 2019) followed by a transformer layer. The principal distinctions between methods lie in the design of the transformer layer. GRAPHGPS utilizes all-pairs attention. EXPHORMER combines GNN with a transformer layer consisting of local neighbor attention and non-local attention via virtual nodes. These virtual nodes are connected to every node in the graph, leading to an approximation of all-node-pairs attention. GOAT (Kong et al., 2023) and LARGEGT (Dwivedi et al., 2023b) removes the GNN component entirely by replacing virtual nodes in EXPHORMER using trainable and clustering-based virtual nodes maintained using a code-book computed by $k$-means clustering and updated through exponentially moving averages.

- **Redundant dependency on GNN:** Hybrid GT such as GRAPHGPS and EXPHORMER utilize a modular architecture of GNN and transformers. While effective, this creates redundant dependence, despite the literature showing that Transformers with positional/structural encodings are theoretically universal approximator of graph-to-graph functions (Yun et al., 2020; Kreuzer et al., 2021a).
- **Limited to either homophilic or heterophilic graphs:** Hybrid architectures inherit the biases of the underlying GNN. If GNN derives the node representation mainly from its local connectivity, then it will propagate homophily biases in the transformer. Similarly, if a GNN is designed for heterophilic graphs that use higher hop nodes, it will disseminate non-homophily biases in the transformer. For example, EXPHORMER and GRAPHGPS use GCN (Kipf & Welling, 2017) as GNN, which leads to their good performance on homophilic graphs (Sen et al., 2008; Yang et al., 2016) but not on heterophilic graphs (Pei et al., 2020; Rozemberczki et al., 2021; Lim et al., 2021). We show this effect empirically in Table 1, where we remove the GCN component to show how their performance improves on heterophilic graphs but reduces on homophilic graphs.
- **Non-scalable or assumptions based:** Dense-attention and global token-based transformers Rampášek et al. (2022); Shirzad et al. (2023) require the full graph in GPU memory, preventing full node batching and limiting scalability. Linear-attention variants (Wu et al., 2023a; Deng et al., 2024; Wu et al., 2023b) improve memory efficiency and increase scalability, but still require graph partitioning for million-node datasets, which breaks all-pairs attention and often sacrifices expressivity due to kernel approximations Deng et al. (2024). Clustering-based virtual node methods Kong et al. (2023); Dwivedi et al. (2023b); Zhu et al. (2024) avoid GNN reliance but require a trainable projection matrix and tuning of the number of virtual nodes.

## 1.2 Contributions

To address the gaps outlined above, we propose Neural Transformer for Attributed Graphs (NEUTAG). We utilize a graph transformation using node features as virtual nodes to design novel sparse graph transformers. We examine the benefits of the aforementioned transformation, including increased graph homophily. We further design a node projection matrix for the proposed transformation and prove that it can approximate dense attention. In summary, NEUTAG offers the following significant advantages over the existing work.

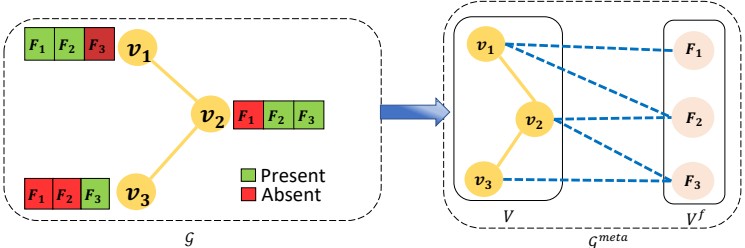

Figure 1: Transformation of input graph $\mathcal{G}$ into its metamorphosis form $\mathcal{G}^{meta}$. In $\mathcal{G}$, the features marked in green are present for the corresponding node. $\mathcal{G}^{meta}$ contains additional edges with all present features.

- **Assumption-free, data agnostic and scalable modeling:** NEUTAG transforms the input graph into a bipartite structure consisting of graph nodes and virtual feature nodes, and sparsifies the all-pair attention into a unified attention mechanism over local neighbors and virtual feature neighbors. This allows NEUTAG to flexibly learn from both homophilic patterns via local structures and heterophilic patterns over non-local neighbors via feature nodes leading to data-agnostic architecture. And, since its virtual nodes are based on features that are derived deterministically from the input attribute space rather than learned through a clustering mechanism, NEUTAG remains fully assumption-free. The resulting sparse-attention mechanism seamlessly facilitates node batching allowing NEUTAG to scale to large-scale graphs.
- **Tighter approximation bounds and theoretical analysis:** We establish the theoretical grounding of the proposed transformation by proving that it increases the connectivity in the graph, and it is equivalent to applying a fixed projection matrix that approximates full-attention, provides reliable and tighter error bounds than clustering based virtual token GT GOAT Kong et al. (2023), which rely on trainable projections. Moreover, we investigate the theoretical and parameter conditions under which the NEUTAG potentially serves as a permutation-equivariant universal approximator of a dense attention layer.
- **Empirical evaluation:** We perform extensive experiments on real-world datasets, including both homophilic and heterophilic datasets, along with a large-scale dataset *snap-patent* containing 2.9 million nodes and 13.9 million edges. We evaluate our proposed method against 12 graph transformer baselines, including 15 variants. We clearly establish that the proposed sparse graph transformer NEUTAG is competitive in both homophilic and heterophilic graphs, as well as in small-scale and large-scale graphs, consistently.

**Paper organization:** Section A in appendix introduces preliminaries on graphs, graph neural networks, transformers, and graph transformers, along with notations and problem formulation. Section 2 presents our proposed methodology: we first describe the graph transformation and its benefits for structural connectivity, then introduce the attention mechanism on the transformed graph and analyze its all-pair attention approximation capabilities. Section 3 details the experimental setup, including datasets, baselines, evaluation metrics, and benchmarks NEUTAG. Finally, Section 4 concludes the paper.

## 2 METHODOLOGY

This section presents two main components. First, we propose a graph transformation that decouples features from nodes into separate nodes, improving structural connectivity and increasing effective homophily, which we formally analyze. Building on this transformation, we introduce a novel attention mechanism that facilitates information flow between graph and feature nodes.

### 2.1 GRAPH TRANSFORMATION

Given a graph $\mathcal{G} = (\mathcal{V}, \mathcal{E}, \mathbf{X})$ and its feature set $F$ as defined in def. 1 in appendix A where $F = \bigcup_{v \in \mathcal{V}} F_v$ is the set of all features in graph $\mathcal{G}$. $F_v$ is a feature set at node $v \in \mathcal{V}$, we convert it to its metamorphosis form $\mathcal{G}^{meta} = (\mathcal{V}^{meta}, \mathcal{E}^{meta}, \mathbf{X})$ as follows. First, we create virtual feature nodes $V^f$ corresponding to the feature set $F$ of graph $\mathcal{G}$ and add these new nodes to $\mathcal{V}$ to create

$\mathcal{V}^{meta}$. Formally,

$$\mathcal{V}^{meta} = \mathcal{V} \cup \mathcal{V}^f : \mathcal{V}^f = \{f \in F\} \tag{1}$$

Next, we retain the original edges $\mathcal{E}$ in $\mathcal{G}^{meta}$ and create the edges that connect original graph nodes $\mathcal{V}^G$ and $\mathcal{V}^f$ as follows.

$$\mathcal{E}^f = \{(v, f) \mid v \in \mathcal{V}, f \in F, \mathbf{X}[v, f] = 1\} \tag{2}$$

We overload the matrix index operation with the access operator $[]$. $\mathcal{E}^f$ represents the set of edges between graph nodes $\mathcal{V}$ and features which are present in corresponding nodes. Thus, edges in $\mathcal{G}^{meta}$ consist of original edges and feature edges. Formally, $\mathcal{E}^{meta} = \mathcal{E} \cup \mathcal{E}^f$.

Figure 1 illustrates this transformation. Since every node in $\mathcal{G}^{meta}$ has two types of neighbors, we specify each possible neighborhood. **1) Graph neighborhood:** nodes from the original graph, $\{\mathcal{N}_v^{\mathcal{G}} = (u \mid (u, v) \in \mathcal{E}\}$, **2) Feature neighbourhood for graph nodes:** feature nodes $\{\mathcal{N}_v^f = (u \mid (u, v) \in \mathcal{E}^f\}$ $\forall v \in \mathcal{V}$, **3) Graph neighbourhood for features nodes:** graph nodes $\{\mathcal{N}_f^{\mathcal{G}} = (u \mid (u, f) \in \mathcal{E}^f\}$ $\forall f \in \mathcal{V}^f$.

## 2.2 TRANSFORMATION BENEFITS

In dense attention, each node ingests information from every other node in a single hop, which requires $\mathcal{O}(N^2)$ computations. The proposed transformation enables nodes to exchange information with *non-local* nodes indirectly via feature nodes, thereby avoiding the explicit computation of pairwise attention scores and reducing computational complexity. This transformation preserves the locality biases and induces connections to distant nodes using feature-connectivity biases. Thus, such transformations increase expressive power by incorporating long-range interactions to learn inductive biases of both homophilic and heterophilic graphs.

**Connectivity analysis:** We assume $D^{\mathcal{G}}$ to be the average graph node degree of graph nodes $\mathcal{V}$, $D^F$ to be the average no. of features for graph nodes $\mathcal{V}$, and $F^{\mathcal{G}}$ be average no. of nodes per feature nodes $\mathcal{V}^f$. We now show that the transformation drastically increases the connectivity in the following theorem.

***Theorem*** 1 (Connectivity of $\mathcal{G}^{meta}$). *Given an input graph $\mathcal{G}$, the average connectivity of a node in L-hop neighbourhood is $\mathcal{O}((D^{\mathcal{G}})^L)$. Let $\mathcal{G}^{meta}$ be a proposed transformed variant of $\mathcal{G}$. The average connectivity in $\mathcal{G}^{meta}$ of the same graph node in L hop significantly increases to $\mathcal{O}((D^{\mathcal{G}})^L + (D^F)^{L/2} * (F^{\mathcal{G}})^{L/2})$.*

**Proof:** See App. B.1. $\qquad\qquad\qquad\qquad\qquad\qquad\qquad\qquad\qquad\qquad\qquad\square$.

Generally $D^F > D^{\mathcal{G}}$ and $F^{\mathcal{G}} \gg D^{\mathcal{G}}$ in real world graphs. This leads to a significant increase in connectivity. The enhanced connectivity leads to quicker reachability to relevant long-range nodes, facilitating distant nodes to become higher personalized rank nodes(Page et al., 1999) for target nodes.

***Corollary*** 1. $\mathcal{G}^{meta}$ facilitates long distances nodes in $\mathcal{G}$ to have better personalized page ranks (PPR).

**Proof:** See App. B.2.

**Higher homophily:** The increased connectivity due to the proposed transformation leads to higher homophily in top PPR nodes in heterophilic graphs. We demonstrate this empirically on two heterophilic graphs, Actor and Chameleon. We compute the top-$K$ nearest PPR nodes for all graph nodes in $\mathcal{G}$ and $\mathcal{G}^{meta}$ and compute the following homophily score per node. $\forall v \in \mathcal{V}$,

$$Homophily_{ppr}^{\mathcal{G}}(v) = \sum_{i=1}^{i=K} \frac{\mid y(v) = y(ppr_i^{\mathcal{G}}(v)) \mid}{K} \tag{3}$$

where $y(v)$ denotes the label of node $v$, $ppr_i(v)^{\mathcal{G}}$ denotes the $i^{th}$ nearest PPR nodes from node $v$ when computed on graph $\mathcal{G}$. We compute the scores for graph nodes on graph $\mathcal{G}$ and $\mathcal{G}^{meta}$. We find that $Homophily_{ppr}$ increases in $\mathcal{G}^{meta}$ for nodes who had lower $Homophily_{ppr}$ in the original graph. To demonstrate this, we define a homophily threshold to identify nodes whose score is less than the threshold in $\mathcal{G}$. Finally, we compare their average homophily score in $\mathcal{G}$ with the same identified nodes in $\mathcal{G}^{meta}$. Their homophily score increases drastically in $G^{meta}$. We note that during


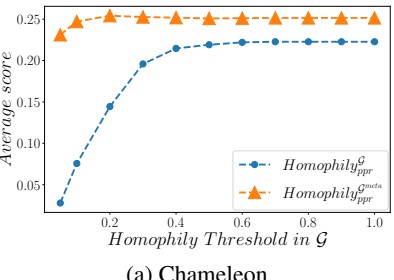
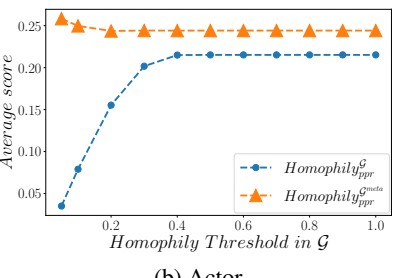

|(a) Chameleon|(b) Actor|

Figure 2: Increase in homophily observed in the transformed graph $\mathcal{G}^{meta}$, while the homophily was low in the original graph $\mathcal{G}$. $K$ is chosen as 50.

top-$K$ ppr node computation in $\mathcal{G}^{meta}$, we pick only graph nodes, disregarding feature nodes $\mathcal{V}^f$. Figure 2 establishes this phenomenon. The difference reduces once the threshold increases more than 0.5.

Next, we discuss our novel attention mechanism on the transformed graph $\mathcal{G}^{meta}$.

## 2.3 GRAPH TRANSFORMER

The main contribution of this work is to propose features as a way of transforming an $N$-dimensional node subspace to a lower $F$-dimensional feature subspace to enable computing attention with feature nodes instead of all graph nodes. This reduces the attention score computation complexity from $\mathcal{O}(N)$ to $\mathcal{O}(F)$ for a given query. We can construct a projection matrix $\mathbf{M} \in \{0,1\}^{N \times F}$ where $\mathbf{M}[i][j] = 1$ when feature $j$ is available in $i^{th}$ node. We now theoretically prove that such a projection matrix exists without considerably degrading the performance compared to dense attention. We utilize the analysis provided in GOAT (Kong et al., 2023). Specifically, we define the following theorem.

***Theorem* 2** (Projection matrix $\mathbf{M}$). *There exists a projection matrix $\mathbf{M} \in \{0,1\}^{N \times F}$ defined as*

$$\mathbf{M}_{ij} = \begin{cases} 1 & \text{if } j^{th} \text{ feature } \in F_i \\ 0 & \text{else} \end{cases}$$

*such that the following holds true for any $\epsilon > 0$, projection matrices $\mathbf{W}_Q, \mathbf{W}_K, \mathbf{W}_V \in \mathcal{R}^{N \times d}$ and node feature matrix $\mathbf{X}$, vector $\mathbf{v} \in \mathbf{X}\mathbf{W}_V$*

$$P(\|\mathbf{A}_{ATN}\mathbf{M}\mathbf{M}^T\mathbf{v} - \mathbf{A}_{ATN}\mathbf{v}\|_F < \epsilon\|\mathbf{A}_{ATN}\mathbf{v}\|_F) > 1 - \mathcal{O}(1/\exp(F)) \quad (4)$$

*And consequently,*

$$P(\|\tilde{\mathbf{A}}_{ATN}\mathbf{M}^T\mathbf{X}\mathbf{W}_v - \mathbf{A}_{ATN}\mathbf{X}\mathbf{W}_V\|_F \le \epsilon\|\mathbf{A}_{ATN}\|_F\|\mathbf{X}\mathbf{W}_V\|_F) > 1 - \mathcal{O}(1/\exp(F)) \quad (5)$$

*where $\mathbf{A}_{ATN}=\text{SOFTMAX}(\mathbf{X}\mathbf{W}_Q(\mathbf{X}\mathbf{W}_K)^T/\sqrt{d})$, $\tilde{\mathbf{A}}_{ATN} = \text{SOFTMAX}((\mathbf{X}\mathbf{W}_Q(\mathbf{X}\mathbf{W}_K)^T/\sqrt{d})\mathbf{M})$ and $\|\dots\|_F$ denotes the Frobenius norm of the matrix.*

**Proof:** See App. B.3. $\qquad\qquad\qquad\qquad\qquad\qquad\qquad\qquad\qquad\qquad\qquad\qquad\qquad\qquad\qquad\quad\square$.

We refer the reader to Appendix B.3 to understand the semantics of the key results 4 and 5. Specifically eq. 5 facilitates to project $\mathbf{X}\mathbf{W}_K, \mathbf{X}\mathbf{W}_V \in \mathcal{R}^{N \times d}$ to $\mathcal{R}^{F \times d}$ by multiplying by $\mathbf{M}$, thus reducing the computation to $\mathcal{O}(N * F * d)$ much less than $\mathcal{O}(N * N * d)$. We also note that, unlike GOAT where the projection matrix $\mathbf{M}$ is learned during training, our approach proposes a fixed and easily derivable projection matrix $\mathbf{M}$, facilitating assumption-free modeling. Additionally, equation 5 provides a tighter approximation error bound than GOAT, whose error scales as $\mathcal{O}(1/\mathcal{V})$. Since $\exp(F) \gg \mathcal{V}$, our bound of $\mathcal{O}(1/\exp(F))$ is substantially tighter. Finally, we now present the NEUTAG architecture.

NEUTAG architecture constructs the attention graph consisting of 4 types of undirected attention edges as shown in fig 3. These attention paths are applied successively across each layer of NEUTAG. The input to the NEUTAG is the node features $\mathbf{X}$, graph topology $\mathcal{E}$, and position encodings (PE) based on random-walk/Laplacian eigenvalue. The input feature matrix is initialized as

$$\mathbf{H}_{\mathcal{V}}^0 = (\mathbf{X} + \mathbf{PE})\mathbf{W} \quad (6)$$

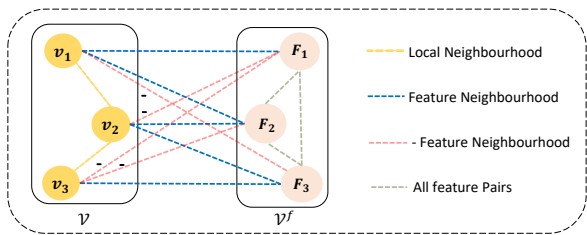

Figure 3: Various attention paths in NEUTAG.

where $\mathbf{H}_{\mathcal{V}}^0 \in \mathcal{R}^{N \times d}$ is a initial representation of graph nodes $\mathcal{V}$. Similarly, for feature nodes $\mathcal{V}^f$, we initialize their embedding to randomly initialized random vectors.

$$\mathbf{H}_{\mathcal{V}^f}^0[f] = w_f \in \mathcal{R}^d \quad \forall f \in \mathcal{V}^f \tag{7}$$

Now, given $\mathbf{H}_{\mathcal{V}}^{l-1} \in \mathcal{R}^{N \times d}$, and $\mathbf{H}_{\mathcal{V}^f}^{l-1} \in \mathcal{R}^{|F| \times d}$, we first compute the following query, key, and value matrices for both graph nodes and feature nodes respectively.

$$\mathbf{Q}_{\mathcal{V}}^l = \mathbf{H}_{\mathcal{V}}^{l-1}\mathbf{W}_1^l, \ \mathbf{K}_{\mathcal{V}}^l = \mathbf{H}_{\mathcal{V}}^{l-1}\mathbf{W}_2^l, \ \mathbf{V}_{\mathcal{V}}^l = \mathbf{H}_{\mathcal{V}}^{l-1}\mathbf{W}_3^l \tag{8}$$

$$\mathbf{Q}_{\mathcal{V}^f}^l = \mathbf{H}_{\mathcal{V}^f}^{l-1}\mathbf{W}_4^l, \ \mathbf{K}_{\mathcal{V}^f}^l = \mathbf{H}_{\mathcal{V}^f}^{l-1}\mathbf{W}_5^l, \ \mathbf{V}_{\mathcal{V}^f}^l = \mathbf{H}_{\mathcal{V}^f}^{l-1}\mathbf{W}_6^l \tag{9}$$

Now we define the following attentions to compute $\mathbf{H}_{\mathcal{V}}^l$ and $\mathbf{H}_{\mathcal{V}^f}^l$. Please note that we provide details using one head for simplicity, but we employ multi-head attention as prevalent in transformers, which entails running attention $H$ times and concatenating these outputs.

**Local neighbourhood attention:** Local neighborhood plays a critical role in node classification accuracy and is commonly used in every sparse transformer, e.g. GRAPHGPS, EXPHORMER, GOAT, LARGEGT, and NAGPHORMER. For each target graph node $v \in \mathcal{V}$, the following vector is computed.

$$\mathbf{H}_{\mathcal{V}:local}^l[v] = \sum_{u \in \mathcal{N}_v^G} \frac{\exp(\mathbf{Q}_{\mathcal{V}}^l[v] * \mathbf{K}_{\mathcal{V}}^l[u]/\sqrt{d})}{\sum_{u' \in \mathcal{N}_v^G} \exp(\mathbf{Q}_{\mathcal{V}}^l[v] * \mathbf{K}_{\mathcal{V}}^l[u']/\sqrt{d})} \mathbf{V}_{\mathcal{V}}^l[u] \tag{10}$$

We apply all-pair attention for each feature node $f \in \mathcal{V}^f$ to compute their local representation at layer $l$.

$$\mathbf{H}_{\mathcal{V}^f:local}^l = \text{SOFTMAX}\left(\frac{\mathbf{Q}_{\mathcal{V}^f}^l(\mathbf{K}_{\mathcal{V}^f}^l)^T}{\sqrt{d}}\right) \mathbf{V}_{\mathcal{V}^f}^l \tag{11}$$

**Attention using feature connections:** The next component utilizes graph nodes to feature nodes connections and vice-versa to learn non-local representations.

For graph nodes $\forall v \in \mathcal{V}$, we use the following.

$$\mathbf{H}_{\mathcal{V}:+}^l[v] = \sum_{f \in \mathcal{N}_v^f} \frac{\exp(\mathbf{Q}_{\mathcal{V}}^l[v] * \mathbf{K}_{\mathcal{V}^f}^l[f]/\sqrt{d})}{\sum_{f' \in \mathcal{N}_v^f} \exp(\mathbf{Q}_{\mathcal{V}}^l[v] * \mathbf{K}_{\mathcal{V}^f}^l[f']/\sqrt{d})} \mathbf{V}_{\mathcal{V}^f}^l[f] \tag{12}$$

For feature nodes $\forall f \in \mathcal{V}^f$, we compute the following.

$$\mathbf{H}_{\mathcal{V}^f:+}^l[f] = \sum_{u \in \mathcal{N}_f^G} \frac{\exp(\mathbf{Q}_{\mathcal{V}^f}^l[f] * \mathbf{K}_{\mathcal{V}}^l[u]/\sqrt{d})}{\sum_{u' \in \mathcal{N}_f^G} \exp(\mathbf{Q}_{\mathcal{V}^f}^l[f] * \mathbf{K}_{\mathcal{V}}^l[u']/\sqrt{d})} \mathbf{V}_{\mathcal{V}}^l[u] \tag{13}$$

**Attention using absent feature connections** Since absent features also provide valuable *exclusion* information in node classification performance, we define the negative feature edges as follows.

$$\mathcal{E}^{f-} = \{(v, f) \mid v \in \mathcal{V}, f \in F, \mathbf{X}[v, f] = 0\} \tag{14}$$

Corresponding to negative edges, we define the negative feature neighborhood for graph nodes $V^{\mathcal{G}}$ as $\{\mathcal{N}_v^- = (u \mid (u, v) \in \mathcal{E}^{f-}\}$. Following, we define another projection matrix $\overline{\mathbf{M}}$ corresponding to these edges, which too satisfies the theorem 2.

$$\overline{\mathbf{M}}_{ij} = \begin{cases} 1 & \text{if } j^{th} \text{ feature} \notin F_i \\ 0 & \text{else} \end{cases}$$

Similar to eq. 10 and 12, we utilize the negative graph connections between graph nodes and feature node connections are utilized as follows. $\forall v \in \mathcal{V}$,

$$\mathbf{H}^l_{\mathcal{V}:-}[v] = \sum_{f \in \mathcal{N}_v^-} \frac{\exp(\mathbf{Q}^l_{\mathcal{V}}[v] * \mathbf{K}^l_{\mathcal{V}f}[f]/\sqrt{d})}{\sum_{f' \in \mathcal{N}_v^-} \exp(\mathbf{Q}^l_{\mathcal{V}}[v] * \mathbf{K}^l_{\mathcal{V}f}[f']/\sqrt{d})} \mathbf{V}^l_{\mathcal{V}f}[f] \tag{15}$$

Since each graph node has multiple negative features, we sample a fixed number of negative feature nodes per node using degree-based sampling for implementation efficiency.

Finally, these local and non-local representations are merged to learn next-layer representations of graph and feature nodes as follows.

$$\mathbf{H}^l_{\mathcal{V}} = \text{UPDATE}^l_1(\mathbf{H}^{l-1}_{\mathcal{V}}, (\text{MLP}^l_1(\mathbf{H}^l_{\mathcal{V}:local} \mid \mathbf{H}^l_{\mathcal{V}:+} \mid \mathbf{H}^l_{\mathcal{V}:-}))) \tag{16}$$

$$\mathbf{H}^l_{\mathcal{V}f} = \text{UPDATE}^l_2(\mathbf{H}^{l-1}_{\mathcal{V}f}, (\text{MLP}(\mathbf{H}^l_{\mathcal{V}f:local} \mid \mathbf{H}^l_{\mathcal{V}f:+}))) \tag{17}$$

Here $\text{UPDATE}^l$ can be a neural net-based functions, e.g. $\text{MLP}$ or skip-connections.

We now show that NEUTAG with positive and negative feature attention paths approximates the Positive Orthogonal Random Projections based sparse-transformer PERFORMER (Choromanski et al., 2020), which kernelizes the softmax operation using Mercer's theorem. We formally define the following theorem.

**Theorem** 3. NEUTAG *can approximate the following self-attention layer of* PERFORMER *applied on $l^{th}$ layer node representation $\mathbf{H}^l$ of $\mathcal{G}$ in **3** proposed attention layers.*

$$\mathbf{h}^{l+1}_i = \frac{\phi(\mathbf{W}_Q \mathbf{h}^l_i)^T \sum_{j=1}^{j=N} \phi(\mathbf{W}_K \mathbf{h}^l_j) \otimes (\mathbf{W}_V \mathbf{h}^l_j)}{\phi(\mathbf{W}_Q \mathbf{h}^l_i) \sum_{k=1}^{k=N} \phi(\mathbf{W}_K \mathbf{h}^l_k)} \tag{18}$$

*Given that a) $\phi$ is a universally approximated kernel function by neural networks and b) each graph node will be connected to at least 1 feature node. Here $\mathbf{h}_i = \mathbf{H}[i]$ is $d$ dimensional vector and $W_Q, W_K, W_V$ are weight matrices. $\phi : \mathcal{R}^d \to \mathcal{R}^m$ is a low dimension random projection based feature mapping.*

**Proof:** See App. B.4. $\square$.

NEUTAG **Mini-Batching:** We request readers to refer appendix section C.1 which contains batching algorithm 1 for running NEUTAG on large-scale graphs.

## 2.4 UNIVERSAL APPROXIMATION CAPABILITIES

Dense transformers have been proven universal approximations of sequence-to-sequence permutation equivariant functions (Yun et al., 2020). The same work further proves transformers are universal approximates of all sequence-to-sequence functions by including position encoding. Further SAN (Kreuzer et al., 2021a) proves that since a graph can be constructed as a sequence on edges or nodes, dense attention-based graph transformers are universal approximates of such sequences within a bound inducing higher expressivity than 1-Weisfeiler Lehman (WL) isomorphism test. Since NEUTAG doesn't utilize all $\mathcal{O}(N^2)$ connections, analyzing its universal approximation capabilities of dense self-attention layer is significant. Formally,

**Theorem** 4. *Given an input graph $\mathcal{G}$, its metamorphosis $\mathcal{G}^{meta}$ and $\mathbf{X} \in \mathcal{R}^{N \times d}$ is a node representation matrix. For following all-pair self attention layer, there exists a NEUTAG's attention layer which is a well permutation equivariant universal approximate with $\mathcal{O}(N^d)$ parameters in $\mathcal{O}(1)$ layers.*

$$\mathbf{H} = \text{SOFTMAX}\left(\frac{(\mathbf{H}^l \mathbf{W}_Q)(\mathbf{H}^l \mathbf{W}_K)^T}{\sqrt{d}}\right) \mathbf{H}^l \mathbf{W}_V \tag{19}$$

*Given every graph node $v \in \mathcal{V}$ is connected to at-least 1 feature node $f \in \mathcal{V}^f$ in $\mathcal{G}^{meta}$.*

**Proof:** Please refer to App. B.5. $\square$.

## 3 EXPERIMENTS

### 3.1 EMPIRICAL EVALUATION

We now evaluate the effectiveness of NEUTAG on node classification tasks across diverse graph datasets and examine its robustness with state-of-the-art graph transformers(GT). We also compare

Table 1: Comparison of NEUTAG against baseline GT on node classification task

| Method | Cora | CiteSeer | Actor | Chameleon | OGBN-Arxiv | OGBN-Arxiv(Year) | Snap-patents |
|---|---|---|---|---|---|---|---|
| GRAPHGPS | $83.65 \pm 2.67$ | $76.25 \pm 1.34$ | $34.30 \pm 0.45$ | $42.87 \pm 1.88$ | OOM | OOM | OOM |
| GRAPHGPS-GNN | $72.47 \pm 1.87$ | $71.59 \pm 2.43$ | $37.10 \pm 1.11$ | $47.36 \pm 2.22$ | OOM | OOM | OOM |
| EXPHORMER | $86.48 \pm 2.15$ | $75.92 \pm 1.88$ | $35.19 \pm 0.94$ | $45.17 \pm 2.56$ | OOM | OOM | OOM |
| EXPHORMER-GNN | $82.35 \pm 1.75$ | $73.01 \pm 1.20$ | $35.44 \pm 0.86$ | $46.97 \pm 0.95$ | OOM | OOM | OOM |
| GRIT | $82.56 \pm 1.80$ | $76.10 \pm 0.67$ | $35.34 \pm 0.76$ | $48.81 \pm 2.26$ | OOM | OOM | OOM |
| GRAPHORMER | $39.45 \pm 10.66$ | OOM | OOM | $26.89 \pm 7.25$ | OOM | OOM | OOM |
| KAA | $82.16 \pm 1.36$ | $71.83 \pm 1.51$ | $34.88 \pm 0.89$ | $45.44 \pm 5.61$ | OOM | OOM | OOM |
| NAGPHORMER | $86.78 \pm 0.77$ | $74.69 \pm 1.06$ | $33.03 \pm 0.75$ | $59.97 \pm 1.72$ | $67.36 \pm 0.12$ | $48.98 \pm 0.23$ | $61.27 \pm 0.13$ |
| GOAT | $84.93 \pm 0.51$ | $76.75 \pm 1.84$ | $\mathbf{37.98 \pm 1.02}$ | $53.28 \pm 2.48$ | $\mathbf{72.17 \pm 0.09}$ | $50.81 \pm 0.36$ | $55.35 \pm 2.24$ |
| LARGEGT | $83.42 \pm 1.21$ | $70.78 \pm 1.62$ | $37.47 \pm 1.62$ | $57.19 \pm 1.89$ | $67.56 \pm 0.20$ | $53.46 \pm 0.78$ | $\mathbf{63.15 \pm 0.002}$ |
| NEUTAG | $\mathbf{87.67 \pm 1.10}$ | $\mathbf{77.68 \pm 1.90}$ | $36.21 \pm 1.2$ | $\mathbf{65.26 \pm 2.43}$ | $70.63 \pm 0.29$ | $\mathbf{53.96 \pm 0.38}$ | $63.00 \pm 0.22$ |

NEUTAG with standard Graph Neural Networks (GNN). Finally, we analyze the importance of feature nodes based on the attention layer via an ablation study. We also analyze the impact of feature sparsity or missing features on NEUTAG's performance in the appendix section D.3.

## 3.2 DATASETS

Table 5 in appendix C.2 summarizes the datasets and their statistics. Cora (Sen et al., 2008), CiteSeer (Yang et al., 2016) and OGBN-Arxiv (Hu et al., 2020) are homophilic datasets while Actor (Pei et al., 2020), Chameleon (Rozemberczki et al., 2021), OGBN-Arxiv(year) (Hu et al., 2020) and Snap-Patents (Lim et al., 2021) are heterophilic datasets. Out of these, Snap-Patents is the largest dataset, having 2.92 million nodes and 13.97 million edges. We use 60%, 20%, and 20% train, validation, and test splits on all datasets for all methods, including baselines. More details on datasets and experiment settings, including hyperparameter values, are given in Appendices C.2 and C.3. The codebase is shared at `https://anonymous.4open.science/r/nutag-7774/`.

## 3.3 BASELINES

We consider state-of-the-art graph transformers for comparison. We evaluate NEUTAG against GRAPHGPS (Rampášek et al., 2022) and its variant GRAPHGPS-GNN where we remove the GNN component to demonstrate the massive decrease in performance and henceforth dependency on MPNN. Similarly, we evaluate against EXPHORMER (Shirzad et al., 2023) and its variant EXPHORMER-GNN as well as GRIT (Ma et al., 2023), KAA (Fang et al., 2025), GRAPHORMER (Ying et al., 2021), NAGPHORMER (Chen et al., 2022b), GOAT (Kong et al., 2023) and LARGEGT (Dwivedi et al., 2023b).

Moreover, we further evaluate NEUTAG against standard and foundational graph neural networks GRAPHSAGE (Hamilton et al., 2017), GAT (Veličković et al., 2018), GIN (Xu et al., 2019), LINKX (Lim et al., 2024) and MIXHOP (Abu-El-Haija et al., 2019). MIXHOP solves the over-smoothing in GNN while LINKX is a strong benchmark method for non-homophilic graphs. There exist multiple complementing techniques which enhance GNN performance, e.g., label-propagation (Huang et al., 2021), adaptive channel mixing (Luan et al., 2024b), gradient-gating (Rusch et al., 2023), data-augmentation (Zhao et al., 2022), (Chowdhury et al., 2023) and knowledge-distillation (Hong et al., 2024). Although these techniques can potentially affect GT architectures, study of their effects is beyond the scope of this paper, and we leave it for future work.

For completeness, we also compare NEUTAG with alternative attention formulations in graph transformers, specifically DIFFORMER (Wu et al., 2023a), SGFORMER (Wu et al., 2023b), POLYNORMER (Deng et al., 2024), and ADVDIFFORMER (Wu et al., 2025). These methods provide equivalent attention formulations that are complementary to sparse graph transformers and can

Table 2: Comparison of NEUTAG with GNN on node classification task

| Method | Cora | CiteSeer | Actor | Chameleon | OGBN-Arxiv | OGBN-Arxiv(Year) | Snap-patents |
|---|---|---|---|---|---|---|---|
| GRAPHSAGE | $87.31 \pm 0.96$ | $76.55 \pm 1.78$ | $34.74 \pm 1.20$ | $48.95 \pm 3.16$ | $61.71 \pm 0.79$ | $46.34 \pm 0.25$ | $49.04 \pm 0.03$ |
| GAT | $86.56 \pm 1.13$ | $76.43 \pm 2.55$ | $30.03 \pm 0.67$ | $44.74 \pm 3.29$ | $62.35 \pm 0.20$ | $44.62 \pm 0.52$ | $36.64 \pm 0.53$ |
| GIN | $84.39 \pm 0.65$ | $75.47 \pm 1.28$ | $26.24 \pm 0.52$ | $32.68 \pm 3.68$ | $59.35 \pm 0.30$ | $46.60 \pm 0.29$ | $47.61 \pm 0.12$ |
| MIXHOP | $87.65 \pm 0.20$ | $76.97 \pm 0.99$ | $35.03 \pm 0.53$ | $47.68 \pm 2.89$ | $62.79 \pm 0.39$ | $44.80 \pm 0.17$ | OOM |
| LINKX | $83.14 \pm 1.62$ | $73.72 \pm 0.14$ | $32.78 \pm 0.17$ | $48.20 \pm 3.31$ | $60.39 \pm 0.32$ | $49.00 \pm 0.39$ | $52.71 \pm 0.19$ |
| NEUTAG | $\mathbf{87.67 \pm 1.10}$ | $\mathbf{77.68 \pm 1.90}$ | $\mathbf{36.21 \pm 1.2}$ | $\mathbf{65.26 \pm 2.43}$ | $70.63 \pm 0.29$ | $\mathbf{53.96 \pm 0.38}$ | $\mathbf{63.00 \pm 0.22}$ |

potentially be integrated with NEUTAG and other baselines GOAT, NAGPHORMER, GRAPHGPS, EXPHORMER, and LARGEGT. Moreover, as discussed in the related work section 1.1, there exists a plethora of work focusing on improving positional and structural encodings, tokenization strategies, or other orthogonal design aspects, which are beyond the scope of this study.

Table 3: Comparison of NEUTAG against alternate attention formulation based GT

| Method | Cora | CiteSeer | Actor | Chameleon | OGBN-Arxiv | OGBN-Arxiv(year) | Snap-patents | Average |
|---|---|---|---|---|---|---|---|---|
| DIFFORMER-s | $87.34 \pm 1.52$ | $77.75 \pm 2.76$ | $31.20 \pm 0.81$ | $57.41 \pm 2.41$ | $40.45 \pm 1.69$ | $36.74 \pm 0.43$ | OOM | NA |
| DIFFORMER-a | $86.01 \pm 2.28$ | $76.70 \pm 1.95$ | $30.79 \pm 1.13$ | $58.07 \pm 1.95$ | OOM | OOM | OOM | NA |
| SGFORMER | $86 \pm 1.76$ | $75.83 \pm 2.28$ | $31.03 \pm 2.99$ | $65.77 \pm 1.68$ | $74.51 \pm 0.31$ | $49.14 \pm 0.34$ | $29.44 \pm 0.84$ | 59.03 |
| POLYNORMER | $87.49 \pm 1.01$ | $75.62 \pm 0.92$ | $37.22 \pm 1.60$ | $67.63 \pm 1.65$ | $74.85 \pm 0.15$ | $52.12 \pm 0.31$ | $34.44 \pm 0.29$ | 61.33 |
| POLYNORMER (-GNN) | $71.66 \pm 0.65$ | $70.90 \pm 1.20$ | $38.16 \pm 1.12$ | $49.39 \pm 1.69$ | $65.84 \pm 0.35$ | $43.04 \pm 0.18$ | $31.10 \pm 0.90$ | 52.87 |
| ADVDIFFORMER | $79.08 \pm 1.30$ | $69.58 \pm 1.95$ | $33.30 \pm 0.89$ | $50.48 \pm 5.13$ | $66.83 \pm 0.13$ | $39.26 \pm 0.58$ | OOM | NA |
| NEUTAG | $87.67 \pm 1.10$ | $77.68 \pm 1.90$ | $36.21 \pm 1.2$ | $65.26 \pm 2.43$ | $70.63 \pm 0.29$ | $53.96 \pm 0.38$ | $63.00 \pm 0.22$ | 64.91 |

## 3.4 RESULT ANALYSIS

**Comparison with Graph Transformers:** Table 1 presents the node classification accuracy of baselines and the proposed model NEUTAG against 7 diverse datasets. The results clearly demonstrate the strong performance of the proposed model NEUTAG with respect to baselines. As we outlined in the introduction, while GRAPHGPS and EXPHORMER perform well on homophilic datasets, their performance drastically deteriorates after removing the GNN component (-GNN), which improves their performance on heterophilic graphs. Moreover, neither method is scalable for large-scale datasets. The recently proposed graph transformers GOAT, NAGPHORMER, and LARGEGT are scalable via global nodes; their performance is inconsistent across all graphs. E.g., GOAT doesn't perform well on Cora, Chameleon, OGBN-Arxiv(year), and Snap-patents while good on CiteSeer, OGBN-Arxiv, and Actor. LARGEGT is overall worse on small-scale graphs but delivers good performance on large-scale datasets. To further validate the effectiveness of NEUTAG, we extend our comparison with these scalable graph transformers in table 7 in the appendix to 5 additional small but challenging heterophilic datasets as defined in the survey Luan et al. (2024a). Table 7 clearly demonstrates the strong performance of NEUTAG across these challenging heterophilic graphs. While no model outperforms all baselines across every dataset due to diverse inductive biases, our proposed assumption-free sparse model NEUTAG adapts well to miscellaneous graphs. NEUTAG delivers stable performance, consistently ranking best or within 1.5% of the top model across all datasets.

**Comparison with alternate attentions:** Table 3 compares NEUTAG against alternative attention-based graph transformers, including DIFFORMER, SGFORMER, and POLYNORMER. DIFFORMER has two variants: DIFFORMER-s and DIFFORMER-a, where the latter employs a non-linear kernel but fails to scale on medium-sized graphs. Neither variant is able to scale to large graphs like snap-patents. Among these baselines, POLYNORMER performs competitively with NEUTAG. However, both SGFORMER and POLYNORMER require partitioning of large graphs into smaller subgraphs, leading to suboptimal results due to limited information exchange in case of snap-patent, unlike sparse GT NEUTAG, which utilizes information exchange between all nodes through its novel propagation framework. We note that these alternative attention mechanisms can be integrated into NEUTAG and other sparse GT baselines to enhance their performance, which we leave for future work.

**Comparison with Graph Neural Networks:** Table 2 presents the performance of NEUTAG against foundational GNN on 7 datasets. Since information propagation in GNN is limited to a few hops, we clearly see that they are competitive with NEUTAG only on homophilic graphs, Cora, and CiteSeer. Consequently, GNN exhibits worse performance than NEUTAG on Chameleon, OGBN-Arxiv(year), and snap-patents, which require long-range interactions. This clearly establishes the necessity for information propagation from distant nodes for an optimal node classification model. Out of baselines, GRAPHSAGE and MIXHOP are consistent performers. Since MIXHOP utilizes a normalized Laplacian matrix, it doesn't scale to large graphs. LINKX designed for heterophilic graphs is competitive on Chameleon and outperforms the rest of the GNN on large-scale snap-patent.

**Ablation Study:** We refer readers to the App. D.1 to analyze the impact of various attention components in NEUTAG.

## 4 CONCLUSION

We introduced NEUTAG, a novel sparse graph transformer that unifies structural and feature information within a single attention mechanism. Unlike prior approaches relying on separate GNN components or virtual nodes, NEUTAG leverages features as global nodes, enabling efficient long-range connectivity. We also provide a theoretical analysis of NEUTAG's capabilities. Finally, experiments on seven real-world datasets demonstrate that NEUTAG achieves competitive and consistent performance across diverse graph types, underlining its generality.

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

## A  PRELIMINARIES

| Symbol | Meaning |
| --- | --- |
| $\mathcal{G}$ | Input graph |
| $\mathcal{V}$ | Set of nodes in $\mathcal{G}$ |
| $\mathcal{E}$ | Set of edges in $\mathcal{G}$ |
| $\mathbf{X}$ | Node feature matrix |
| $F$ | Set of features in $\mathcal{G}$ |
| $\mathcal{G}^{meta}$ | Transformed input graph |
| $\mathcal{V}^f$ | Set of features as virtual node in $\mathcal{G}^{meta}$ |
| $\mathcal{E}^f$ | Set of edges between nodes and respective features $\mathcal{G}^{meta}$ |
| $\mathcal{E}^{f-}$ | Set of edges between nodes and absent features nodes in $\mathcal{G}^{meta}$ |
| $\mathcal{V}^{meta}$ | Set of nodes in $\mathcal{G}^{meta}$ |
| $\mathcal{E}^{meta}$ | Set of edges in $\mathcal{G}^{meta}$ |
| $D^{\mathcal{G}}$ | Average node degree excluding feature nodes in $\mathcal{G}^{meta}$ |
| $D^F$ | Average node degree excluding graph nodes in $\mathcal{G}^{meta}$ |
| $y(v)$ | Label of node $v$ |
| $ppr_i^{\mathcal{G}}(v)$ | $i^{th}$ nearest node from node $v$ sorted using personalized page rank score |
| $\mathbf{M}$ | Projection matrix |
| $\mathbf{h}_i^l$ | Embedding of node $i$ at $l^{th}$ layer |
| $\mathbf{H}_{\mathcal{V}}^l$ | Embedding matrix of graph nodes $\mathcal{V}$ at layer $l$ |
| $\mathbf{H}_{\mathcal{V}^f}^l$ | Embedding matrix of feature nodes $\mathcal{V}^f$ at layer $l$ |
| $\mathbf{W}$ | Learnable weight matrices |
| $\mathbf{Q}_{\mathcal{V}}^l$ | Query matrix at layer $l$ for graph nodes $\mathcal{V}$ |
| $\mathbf{K}_{\mathcal{V}}^l$ | Key matrix at layer $l$ for graph nodes $\mathcal{V}$ |
| $\mathbf{V}_{\mathcal{V}}^l$ | Value matrix at layer $l$ for graph nodes $\mathcal{V}$ |
| $\mathbf{Q}_{\mathcal{V}^f}^l$ | Query matrix at layer $l$ for feature nodes $\mathcal{V}^f$ |
| $\mathbf{K}_{\mathcal{V}^f}^l$ | Key matrix at layer $l$ for feature nodes $\mathcal{V}^f$ |
| $\mathbf{V}_{\mathcal{V}^f}^l$ | Value matrix at layer $l$ for feature nodes $\mathcal{V}^f$ |
| $\mathcal{N}_v^{\mathcal{G}}$ | Set of neighbors excluding feature nodes of node $v$ in $\mathcal{G}^{meta}$ |
| $\mathcal{N}_v^f$ | Set of neighbors consisting of only feature nodes of node $v$ in $\mathcal{G}^{meta}$ |
| $\mathcal{N}_f^{\mathcal{G}}$ | Set of neighbors excluding feature nodes of a feature node $f$ in $\mathcal{G}^{meta}$ |

Table 4: Notations and their definition

***Definition*** 1 (Graph). *A graph is defined as $\mathcal{G} = (\mathcal{V}, \mathcal{E}, \mathbf{X})$ over node and edge sets $\mathcal{V}$ and $\mathcal{E} = \{(u, v) \mid u, v \in \mathcal{V}\}$ respectively where $|\mathcal{V}| = N$ and $|\mathcal{E}| = M$. Edge set is also represented using an adjacency matrix $\mathcal{A} \in \{0, 1\}^{N \times N}$. $\mathbf{X} \in \{0, 1\}^{N \times |F|}$ is a node feature matrix where $F = \bigcup_{v \in \mathcal{V}} F_v$ is the set of all features in graph $\mathcal{G}$. $F_v$ is a feature set at node $v$.*

***Problem*** 1 (Graph transformer for node classification).

**Input:** *Given a graph $\mathcal{G}$ (Def. 1), let $Y : \mathcal{V} \to \mathbb{R}$ be a hidden function that maps a node to a real number. $Y(v)$ is known to us only for the subset $\mathcal{V}_l \subset \mathcal{V}$ and may model some downstream tasks such as node classification or link prediction.*

**Goal:** *Learn parameters $\Theta$ of a Transformer based graph neural network, denoted as $\mathrm{GT}_{\Theta}$, that predicts $Y(v)$, $\forall v \in \mathcal{V}_l$ accurately.*

We now introduce preliminaries of Graph Neural Networks, Transformers, and Graph Transformers for node classification tasks.

## A.1 GRAPH

GNN also known as message-passing neural networks, as each node exchanges messages from its neighbors to compute its representation. These representations are utilized in downstream tasks such as node classification and link prediction. Though there exists more specialized GNN for the link-prediction task that utilizes link-based features instead of node-level, those are out of scope in this work. State-of-the-art GNN (Xu et al., 2019; Veličković et al., 2018; Hamilton et al., 2017) for node classification tasks follows the following framework. Assuming $\mathbf{x}_v \in \mathcal{R}^{|F|}$ as feature vector for node $v$, $0^{th}$ layer embedding is defined as:

$$\mathbf{h}_v^0 = \mathbf{x}_v \ \forall v \in \mathcal{V} \tag{20}$$

Next, $l^{th}$ layer representation is computed using nodes' neighbourhood $\mathcal{N}_v = \{u \mid (u,v) \in \mathcal{E}\} \ \forall \ v \in \mathcal{V}$ as follows.

$$\mathbf{m}_v^l = \text{MSG}(\mathbf{h}_u^{l-1}, \mathbf{h}_v^{l-1}) \forall u \in \mathcal{N}_v \tag{21}$$

Messages are computed from each neighbor using the previous layer information. This information is then aggregated at each node as follows.

$$\overline{\mathbf{m}}_v = \text{AGGREGATE}^l(\{\!\{\mathbf{m}_v^l(u), \forall u \in \mathcal{N}_v\}\!\}) \tag{22}$$

$\{\!\{\ldots\}\!\}$ is a multi-set as the same message can arrive from multiple neighbors. Multi-set allows multiple instances of the same element achieving improved expressivity, highlighted in (Xu et al., 2019). Finally, the aggregated message and previous layer $l-1$ representation are combined to compute the $l^{th}$ layer representation as follows.

$$\mathbf{h}_v^l = \text{COMBINE}(\mathbf{h}_v^{l-1}, \overline{\mathbf{m}}_v) \tag{23}$$

where MSG, AGGREGATE and COMBINE are non-neural functions like SUM, AVERAGE or MAX-POOL or neural networks based learnable functions like *mlp*, *attention* (Vaswani et al., 2017) and *recurrent neural networks* eg. GRU (Dey & Salem, 2017). To achieve $L$ hop deeper GNN, equations 21, 22 and 23 are applied $L$ times successively to compute $\mathbf{h}_v^L$. This representation is utilized for node classification tasks. GNN are limited in modeling long-range dependencies as increasing the number of layers leads to *over-smoothing*(Oono & Suzuki, 2020; Chen et al., 2020) where node embeddings become approximately similar at every node. Graph Transformers solves this by introducing the mechanism of each node attending to all other nodes as follows.

## A.2 TRANSFORMERS

First, we define the transformer neural nets, the key components of graph transformers. Given a graph $\mathcal{G} = (\mathcal{V}, \mathcal{E}, \mathbf{X})$ and ignoring topological connections $\mathcal{E}$, contextualized node representations $\mathbf{H}^L$ are computed using self-attention. The representations are first initialized using node features.

$$\mathbf{H}^0 = \mathbf{X}\mathbf{W} \tag{24}$$

where $\mathbf{W}$ is the trainable weight matrix. Next, below equations 25 and 26 are repeated for $l \in [1 \ldots L]$ as follows.

$$\mathbf{H}^l = \Big\|_{h=1}^{h=H} \text{SOFTMAX}\left(\frac{(\mathbf{H}^{l-1}\mathbf{W}_Q^h)(\mathbf{H}^{l-1}\mathbf{W}_K^h)^T}{\sqrt{d}}\right)\mathbf{H}^{l-1}\mathbf{W}_V^h \tag{25}$$

$$\mathbf{H}^l = \text{NORM}(\mathbf{H}^{l-1} + \text{FFN}(\mathbf{H}^l)) \tag{26}$$

where NORM is either batch-norm (Ioffe & Szegedy, 2015) or layer-norm (Ba et al., 2016), FFN is feed-forward neural network. $\mathbf{W}_Q$, $\mathbf{W}_K$ and $\mathbf{W}_V$ are projection matrix $\in \mathcal{R}^{d \times d_k}$. $H$ is a number of heads, and the transformer concatenates these multiple heads, facilitating diverse attention coefficients. This is also known as *Multi-head Attention*. This architecture uses skip-connections and activation norm strategies required in deeper neural networks He et al. (2016).

### A.3 GRAPH TRANSFORMERS (GT)

Now, we discuss graph transformers, which incorporate graph topology $\mathcal{E}$ into the attention mechanism. Foremost, node attributes are combined with position encodings, e.g., Random walk-based encoding (Dwivedi et al., 2022a) and Laplacian eigenvalues (Dwivedi et al., 2023a) denoted as **PE** matrix.

$$\mathbf{H}^0 = (\mathbf{X} + \mathbf{PE})\mathbf{W} \tag{27}$$

Next, assuming we have node presentation for $l-1$ layer, it is passed to the GNN layer along with the transformer layer to compute $l^{th}$ layer representation as follows.

$$\mathbf{H}_{gnn}^l = \text{GNN}(\mathbf{H}^{l-1}, \mathcal{E}) \tag{28}$$

$$\mathbf{H}_T^l = \text{TRANSFORMER}(\mathbf{H}^{l-1}, \mathcal{E}) \tag{29}$$

where GNN is any graph neural network described earlier. TRANSFORMER layer can be defined as in current literature (Rampášek et al., 2022; Shirzad et al., 2023; Kong et al., 2023; Dwivedi et al., 2023b). Finally, the representation computed using GNN and TRANSFORMER are combined to learn $l^{th}$ layer representation.

$$\mathbf{H}^l = \text{FFN}(\mathbf{H}_{gnn}^l, \mathbf{H}_T^l) \tag{30}$$

As highlighted in the Introduction, dependencies on GNN in the existing transformer lead to a wide range of issues, including applicability on either homophilic or heterophilic graphs, along with scalability issues due to dense attention. We now explain our methodology, which is GNN independent, scalable, and expressive.

## B PROOF OF THEOREMS

### B.1 CONNECTIVITY ANALYSIS OF $\mathcal{G}^{meta}$: PROOF OF THEOREM 1

**Proof:** First, we clarify the notation. We assume $D^{\mathcal{G}}$ to be the average graph node degree of graph nodes $\mathcal{V}$, $D^F$ to be the average no. of features for graph nodes $\mathcal{V}$, and $F^{\mathcal{G}}$ be average no. of nodes per feature nodes $\mathcal{V}^f$. With these, we derive the approximate $L$-hop neighbors of graph nodes $\mathcal{V}$ in $\mathcal{G}^{meta}$. First, we define the number of 1-hop neighbors of the graph and feature nodes where $\#Nbrs(\mathcal{V}, l)$ signifies the order of $l$-hop neighbors of graph nodes $\mathcal{V}$ and $\#Nbrs(\mathcal{V}^f, l)$ is the order of number of $l$-hop neighbors of feature nodes $\mathcal{V}^f$.

$$\#Nbrs(\mathcal{V}, 1) = O(D^{\mathcal{G}} + D^F), \quad \#Nbrs(\mathcal{V}^f, 1) = O(F^{\mathcal{G}}) \tag{31}$$

As each $D^F$ feature node will connect to graph nodes connected to it, and each graph node $F^{\mathcal{G}}$ and $D^{\mathcal{G}}$ will be connected to its neighbors and feature nodes, applying this for 2 hops,

$$\begin{aligned} \#Nbrs(\mathcal{V}, 2) &= O(D^{\mathcal{G}} * (D^{\mathcal{G}} + D^F) + D^F * F^{\mathcal{G}}), \\ \#Nbrs(\mathcal{V}^f, 2) &= O(F^{\mathcal{G}} * (D^{\mathcal{G}} + D^F)) \end{aligned} \tag{32}$$

$$\#Nbrs(\mathcal{V}, 3) = O(D^{\mathcal{G}} * (D^{\mathcal{G}} * (D^{\mathcal{G}} + D^F) + D^F * F^{\mathcal{G}}) + D^F * F^{\mathcal{G}} * (D^{\mathcal{G}} + D^F)) \tag{33}$$

$$\begin{aligned} \#Nbrs(\mathcal{V}, 4) =& O(D^{\mathcal{G}} * (D^{\mathcal{G}} * (D^{\mathcal{G}} * (D^{\mathcal{G}} + D^F) + D^F * F^{\mathcal{G}}) + D^F * F^{\mathcal{G}} * (D^{\mathcal{G}} + D^F)) \\ &+ D^F * F^{\mathcal{G}} * (D^{\mathcal{G}} * (D^{\mathcal{G}} + D^F) + D^F * F^{\mathcal{G}})) \end{aligned} \tag{34}$$

While the closed form for $L$-hop neighbor is not feasible, we re-write $\#Nbrs(\mathcal{V}, 4)$ using its recursive nature,

$$\begin{aligned} \#Nbrs(\mathcal{V}, 4) =& O(D^{\mathcal{G}} * (D^{\mathcal{G}} * \#Nbrs(\mathcal{V}, 2)) + D^F * F^{\mathcal{G}} D^{\mathcal{G}} + (D^F)^2 F^{\mathcal{G}} \\ &+ D^F * F^{\mathcal{G}} * \#Nbrs(\mathcal{V}, 2) + D^F * F^{\mathcal{G}}) \end{aligned} \tag{35}$$

where $\#Nbrs(\mathcal{V}, 2)$ contains $D^F * F^{\mathcal{G}}$. Thus, we see that $D^{\mathcal{G}}$ grows with the power of the number of hops $L$, and $D^F * F^{\mathcal{G}}$ multiplies after every other hop. Consequently, we approximate $Nbrs(\mathcal{V}, L)$ as

$$\#Nbrs(\mathcal{V}, L) \geq O((D^{\mathcal{G}})^L + (D^F)^{L/2} * (F^{\mathcal{G}})^{L/2}) \tag{36}$$

$\square$.

## B.2  PROOF OF COROLLARY 1

**Proof:** We can see this via the personalized page-rank (*PPR*) equation $\pi(v) = (1 - \alpha)\tilde{\mathcal{A}}\pi(v) + \alpha\mathbf{i}_v$ where $\tilde{\mathcal{A}}$ is self-loop added normalized adjacency matrix and $\mathbf{i}_v$ is an indicator vector with indices except corresponding to node $v$ are filled with 0, facilitating transportation to target node $v$ in restart. The *PPR* equation can be rewritten as $\pi(v) = \alpha * \mathbf{i}_v(\mathbf{I} - (1 - \alpha)\tilde{\mathcal{A}})^{-1} = \alpha * \sum_{r=0}^{i=\infty}(1 - \alpha)^r \tilde{\mathcal{A}}^r \mathbf{i}_v$ under conditions defined in PAGERANK-NIBBLE (Andersen et al., 2006). I is an identity matrix the same size as $\tilde{\mathcal{A}}$. As seen in the equation, the nearest nodes have a higher weightage of $(1 - \alpha)$, which decays exponentially $(1 - \alpha)^r$ with longer hops where $r$ is the shortest distance from the target node. Thus, the proposed transformation adds connection in the $\mathcal{G}$, facilitating interaction with long-range nodes co-occurring within multiple features.  $\square$.

## B.3  PROJECTION MATRIX M: PROOF OF THEOREM 2

**Proof:** Our proof is derived from proof of proposition 1 and theorem 1 in GOAT (Kong et al., 2023). We write the following Johnson-Lindenstrauss lemma (Johnson & Lindenstrauss, 1984).

**Lemma B.1** (Johnson-Lindenstrauss(JL) Lemma (Johnson & Lindenstrauss, 1984))**.** *For any integer $d > 0$, any $0 < \epsilon, \delta <= 1/2$, there exists a probability distribution $\mathbb{P}$ on $k \times d$ real matrices for $k = \mathcal{O}(\epsilon^{-2}\log(1/\delta))$ such that for any $\mathbf{x} \in \mathcal{R}^d$, following holds*

$$\mathbb{P}_M((1 - \epsilon)\|\mathbf{x}\|_2 \leq \|\mathbf{M}\mathbf{x}\|_2 \leq (1 + \epsilon)\|\mathbf{x}\|_2) > 1 - \delta \tag{37}$$

Following the proof in GOAT, we get to the following.

$$P(\|\mathbf{A}_{ATN}\mathbf{M}\mathbf{M}^T\mathbf{v}^T - \mathbf{A}_{ATN}\mathbf{v}^T\|_F \leq \epsilon\|\mathbf{A}_{ATN}\mathbf{v}^T\|_F) \\ > 1 - 2N\delta \tag{38}$$

Since $F << N$, we can rewrite the above as follows.

$$P(\|\mathbf{A}_{ATN}\mathbf{M}\mathbf{M}^T\mathbf{v}^T - \mathbf{A}_{ATN}\mathbf{v}^T\|_F \leq \epsilon\|\mathbf{A}_{ATN}\mathbf{v}^T\|_F) \\ > 1 - 2F\delta \tag{39}$$

Now we choose $\delta = \mathcal{O}(1/(F\exp(F)))$ honoring the lemma B.1, leading us to

$$P(\|\mathbf{A}_{ATN}\mathbf{M}\mathbf{M}^T\mathbf{v}^T - \mathbf{A}_{ATN}\mathbf{v}^T\|_F \leq \epsilon\|\mathbf{A}_{ATN}\mathbf{v}^T\|_F) \\ > 1 - \mathcal{O}(1/\exp(F)) \tag{40}$$

and $k = \mathcal{O}(\epsilon^{-2}(logF + F)) \approx \mathcal{O}(F)$ which shows us that there exist $\mathcal{R}^{N \times F}$ projection matrices $M$ which are super-set of $\{0, 1\}^{N \times F}$ as defined in the theorem. This completes our first part of the proof.

We have investigated projecting $\mathbf{A}_{ATN}$ to a lower dimension, but it is still computationally expensive as computing $\mathbf{A}$ is $\mathcal{O}(N \times N)$ operation. Next, we explore the moving projection matrix $\mathbf{M}$ inside the softmax $\mathbf{A}_{ATN}$ and analyze its approximation to $\mathbf{A}_{ATN}\mathbf{v}$. Since SOFTMAX$(\mathbf{A}_{ATN}) = \exp(\mathbf{A}_{ATN})\mathbf{D}_{\mathbf{A}_{ATN}}^{-1}$ where $D_{\mathbf{A}_{ATN}}^{-1}$ is diagonal matrix of $\exp(\mathbf{A}_{ATN})$. Following LINFORMER (Wang et al., 2020) and GOAT (Kong et al., 2023), we prove

$$P(\|\exp(\mathbf{A}\mathbf{M})\mathbf{M}^T\mathbf{V} - \exp(\mathbf{A})\mathbf{V}\| \leq \epsilon\|\exp(\mathbf{A})\|_F\|\mathbf{V}\|_F) \\ > 1 - O(1/\exp(F)) \tag{41}$$

where $\mathbf{A}_{ATN} = \text{SOFTMAX}(\mathbf{A})$, $\mathbf{A} = (\mathbf{X}\mathbf{W}_Q(\mathbf{X}\mathbf{W}_K)^T/\sqrt{d})$ and $\mathbf{V} = \mathbf{X}\mathbf{W}_V$. Subsequently, we break the eq. 41 using triangle inequality as follows.

$$\| \exp(\mathbf{A}\mathbf{M})\mathbf{M}^T\mathbf{V} - \exp(\mathbf{A})\mathbf{V} \|_F$$

$$\leq \| \exp(\mathbf{A}\mathbf{M})\mathbf{M}^T\mathbf{V} - \exp(\mathbf{A})\mathbf{M}\mathbf{M}^T\mathbf{V} \|_F$$

$$+ \| \exp(\mathbf{A})\mathbf{M}\mathbf{M}^T\mathbf{V} - \exp(\mathbf{A})\mathbf{V} \|_F \tag{42}$$

$$\overset{a}{\leq} \| \mathbf{V} \|_F \| \exp(\mathbf{A}\mathbf{M})\mathbf{M}^T - \exp(\mathbf{A})\mathbf{M}\mathbf{M}^T \|_F$$

$$+ \| \exp(\mathbf{A})\mathbf{M}\mathbf{M}^T\mathbf{V} - \exp(\mathbf{A})\mathbf{V} \|_F \tag{43}$$

$$\overset{b}{\leq} \| \mathbf{V} \|_F(1+\epsilon)\| \exp(\mathbf{A}\mathbf{M}) - \exp(\mathbf{A})\mathbf{M} \|_F$$

$$+ \epsilon\| \exp(\mathbf{A}) \|_F \| \mathbf{V} \|_F \tag{44}$$

$$\overset{c}{\leq} \epsilon\| \mathbf{V} \|_F \| \exp(\mathbf{A}) \|_F + \epsilon\| \exp(\mathbf{A}) \|_F \| \mathbf{V} \|_F \tag{45}$$

$$\leq \epsilon\| \exp(\mathbf{A}) \|_F \| \mathbf{V} \|_F > 1 - \mathcal{O}(1/\exp(F)) \tag{46}$$

Step - $a$ employs Cauchy inequality, step- $b$ utilizes JL lemma, and step - $c$ utilizes Lipschitz continuity in a compact region. $\qquad\square$

## B.4 NEUTAG IS AN APPROXIMATION OF A SPARSE TRANSFORMER PERFORMER: PROOF OF THEOREM 3

**Proof:** We use the similar strategy outlined in (Cai et al., 2023) which showed that global nodes can approximate PERFORMER. First, we rewrite the equation 18 as follows to simplify the analysis.

$$\mathbf{h}_i^{l+1} = \frac{\phi_1(\mathbf{h}_i^l)^T \sum_{j=1}^{j=N} \phi_2(\mathbf{h}_j^l) \otimes \phi_3(\mathbf{h}_j^l)}{\phi_1(\mathbf{h}_i^l)^T \sum_{o=1}^{o=N} \phi_2(\mathbf{h}_o^l)} \tag{47}$$

where we commutate $\phi$ and weights $\mathbf{W}_Q, \mathbf{W}_K, \mathbf{W}_V$ as $\phi_1(\mathbf{h}) = \phi(\mathbf{W}_Q\mathbf{h})$, $\phi_2(\mathbf{h}) = \phi(\mathbf{W}_K\mathbf{h})$ and $\phi_3(\mathbf{h}) = \mathbf{W}_V\mathbf{h}$ as we will use universal approximation capability of a neural network specifically Multi-Layer Perceptron(MLP). Intuitively, feature nodes can facilitate approximation of both summations $\sum_{j=1}^{j=N} \phi_2(\mathbf{h}_j) \otimes \phi_3(\mathbf{h}_j)$ and $\sum_{o=1}^{o=N} \phi_2(\mathbf{h}_o)$ as each graph nodes is attended by at least 1 feature node. To prove theorem 3, let us assume that in 16, $\mathbf{H}_{\mathcal{V}:local}$ is ignored by UPDATE$_1^l$ and in 17 $\mathbf{H}_{\mathcal{V}f:local}$ is ignored by UPDATE$_2^l$. Since we are analyzing the approximation of layer $l$ of PERFORMER, we will subdivide layer $l$ of NEUTAG as $(l1, l2\ldots)$. Now, at layer $l$, we are provided with $\mathbf{h}_v \forall v \in \mathcal{V}$, let us assume that $\mathbf{h}_f = [\ldots, \mathbb{I}_f] \; \forall f \in \mathcal{V}^f$ where $\mathbb{I}_f$ is a one-hot indicator vector with all zeros except $f^{th}$ index which is equal to 1. We keep the $\mathbb{I}_f$ from layer $l = 0$ itself to facilitate the feature degree calculation at graph nodes. Next, using equations 12 by learning equal attention coefficients for all present feature nodes, and eq. 17 where it omits $\mathbf{H}_{\mathcal{V}f:local}$ and $\mathbf{H}_{\mathcal{V}f:+}$ and learn $\phi_2(\mathbf{h}_v^l)$, $\phi_2(\mathbf{h}_v^l) \otimes \phi_3(\mathbf{h}_v^l)$ and feature degree $d^F(v)$. After this step, $\mathbf{h}_v^{l1}$ can be approximated either as **1)** $[\phi_2(\mathbf{h}_v^l), (\phi_2(\mathbf{h}_v^l) \otimes \phi_3(\mathbf{h}_v^l))_{flattened}, d^F(v)]$ or **2)** $[\phi_2(\mathbf{h}_v^l)/d^F(v), (\phi_2(\mathbf{h}_v^l) \otimes \phi_3(\mathbf{h}_v^l))_{flattened}/d^F(v)]$. Now at layer $l2$, in the first case, each feature node $f$ using the equation 13 learn attention coefficients equal to $\frac{1}{d^F(v)}$ for $v \in \mathcal{N}_f^{\mathcal{G}}$ and computes vector $[\sum_{v \in \mathcal{N}_f^{\mathcal{G}}} \phi_2(\mathbf{h}_v^l)/d_v^F, \sum_{v \in \mathcal{N}_f^{\mathcal{G}}}(\phi_2(\mathbf{h}_v^l) \otimes \phi_3(\mathbf{h}_v^l))_{flattened}/d_v^F, \mathbb{I}_f]$. $\mathbb{I}_f$ is crucial to facilitate the same operations for approximating the next layer of PERFORMER. In the 2nd case, equal attention coefficients are learned, and an identical feature representation is learned. Finally at layer $l3$, each graph node compute the sums $[\sum_{v=1}^{v=N} \phi_2(\mathbf{h}_v^l), \sum_{v=1}^{v=N}(\phi_2(\mathbf{h}_v^l) \otimes \phi_3(\mathbf{h}_v^l))_{flattened}]$ and approximate the required PERFORMER eq. 47 using via equation 12,15 and eq. 16. Consequently, the number of parameters required by each layer NEUTAG to approximate PERFORMER layer is constant with respect to the number of nodes $N$. $\qquad\square$

## B.5 UNIVERSAL APPROXIMATION ANALYSIS OF NEUTAG

**Proof:** Similar to (Cai et al., 2023), we first prove the connection between NEUTAG and DEEPSETS (Zaheer et al., 2017), which is a universal approximator of sequence-to-sequence permutation equivariant functions. Since eq. 19 is also a permutation equivalent function, DEEPSETS can approximate

eq. 19, thus establishing NEUTAG's capability of approximating eq. 19. Formally, we define the following lemma.

**Definition** 2 (DeepSets (Zaheer et al., 2017)). *Each layer of* DEEPSETS *is defined as follows.*

$$\mathbf{H}^{l+1} = \sigma(\mathbf{H}^l \mathbf{W}_1 + \frac{1}{N} \mathbf{1} \mathbf{1}^T \mathbf{H}^l \mathbf{W}_2) \tag{48}$$

*where $\sigma$ is a non-linearity activation function, $\mathbf{H}^l$ is output of previous layer, $\mathbf{1} = [1, 1, \ldots]^T$ is $N$ dimensional vector and $\mathbf{W}_1$, $\mathbf{W}_2$ are weights.*

$\frac{1}{N} \mathbf{1} \mathbf{1}^T$ calculates the average of input $\mathbf{H}^l \mathbf{W}_2$ which is added to $\mathbf{H}^l \mathbf{W}_1$. This function can easily be verified as a permutation equivariant function, as node reordering will only permute the output. Now, we formally write the following lemma of (Segol & Lipman, 2019).

**Lemma B.2** ((Segol & Lipman, 2019)). DEEPSETS *with $\mathcal{O}(1)$ layers and $\mathcal{O}(N^d)$ parameters per layer is a universal for permutation equivariant sequence to sequence functions.*

Thus, DEEPSETS can approximate self-attention layer eq. 19. Moreover, similar to proof of theorem 3, both operations **a)** calculating average of node embeddings and **b)** adding the calculated average to node representation and applying $\sigma$ can be simulated by 3 layer NEUTAG. This concludes our analysis.                                           □.

## C    ADDITIONAL EXPERIMENTAL DETAILS

### C.1    MINI-BATCHING OF NEUTAG

The proposed framework is applicable for **a) Small graphs** by running forward pass on entire graph $\mathcal{G}^{meta}$ and **b) Large-scale graphs** by offline sampling $L$ layer directed sub-graphs from feature nodes $\mathcal{V}^f$ to graph nodes $\mathcal{V}$ from $\mathcal{G}^{meta}$, as these sub-graphs will be common among all batches of graph nodes $\mathcal{V}$ and run NEUTAG on such constructed batches and back-propagate. Algorithm 1 summarizes the creation of a mini-batch for a large-scale graph for NEUTAG training and inference. Specifically, for each feature node $v^f \in \mathcal{V}^f$, graph nodes are sampled in line 4. Corresponding these graph nodes, $L$ hop sub-graph is sampled in original node space $\mathcal{V}$. Please note that data from lines 3-12 can be cached across multiple batches and performed offline. Finally, a batch of nodes is sampled from the input graph, and the corresponding L-HOP subgraph is sampled. Finally, feature edges are added to correspond to these sampled original graph nodes.

### C.2    DATASETS

Cora (Sen et al., 2008) and CiteSeer(Yang et al., 2016) are co-citation graphs where nodes are papers, and their features are bag-of-words of text. The task is to predict the research category of the node. Actor (Pei et al., 2020) is a co-occurrence graph of actors on the same wiki page. Node attributes are bag-of-words from the actor's Wikipedia page, and their labels are actor categories. Chameleon (Rozemberczki et al., 2021) is a graph of hyperlinks between English wiki pages, attributes are nouns, and the label is the binned average monthly traffic on the page. Snap-patents (Lim et al., 2021) is a large-scale co-citation graph of U.S. utility patents where attributes are patent metadata and class label is the time at which the patent was granted, binned in 5 classes. OGBN-Arxiv (Hu et al., 2020) is also a co-citation network where features are 128-dimension embeddings of title and abstract, and the label is the research category. OGBN-Arxiv(Year) is the same graph, but the label is the year of publication, and it is a non-homophilic graph. $\mathbf{H}_{edge}$ (Zhu et al., 2020) in table 5 denotes the edge homophily of a graph.

### C.3    HARDWARE DETAILS

We have performed experiments on an Intel Xeon 6248 processor with a Tesla V-100 GPU with 32GB GPU memory and Ubuntu 18.04. Train, validate, and test data split of 60%, 20%, and 20%, which are generated randomly for every run. We perform 5 runs of every experiment to report the mean and standard deviation. We use $4 - 6$ layer NEUTAG for small graphs and 2 layer for large graphs. We use Adam optimizer to train the model using a learning rate of 0.00001 and choose the best model based on validation loss. For all methods, including baselines and NEUTAG, we apply laplacian position

---

**Algorithm 1** NEUTAG Mini-batching algorithm

---

**Require:** $\mathcal{G}^{meta} = (\mathcal{V}^{meta} = \mathcal{V} \cup \mathcal{V}^f, \mathcal{E}^{meta} = \mathcal{E} \cup \mathcal{E}^f)$, # of layers $L$, Node feature matrix $\mathbf{X}$

**Ensure:** Sampled mini batch $\mathcal{G}^{meta'}$

1: $\mathcal{V}^{meta'} = \{\}$
2: $\mathcal{E}^{meta'} = \{\}$
   {Sample a $L$ hop subgraph from each feature node. This step can be cached as well as it will remain same across all batches}
3: **for** $v^f \in \mathcal{V}^f$ **do**
4:    $\mathcal{V}^{sampled} \sim 1\text{-HOP}(\mathcal{G} = (\mathcal{V}, \mathcal{E}^f), v^f)$
5:    $(\mathcal{V}^L, \mathcal{E}^L) \sim L\text{-HOP}(\mathcal{G} = (\mathcal{V}, \mathcal{E}), \mathcal{V}^{sampled})$
6:    $\mathcal{V}^{meta'} \leftarrow \mathcal{V}^{meta'} \cup \mathcal{V}^L$
7:    $\mathcal{E}^{meta'} \leftarrow \mathcal{E}^{meta'} \cup \mathcal{E}^L$
8:    **for** $v \in \mathcal{V}^{sampled}$ **do**
9:      $\mathcal{E}^{meta'} \leftarrow \mathcal{E}^{meta'} \cup (v, v^f)$
10:    **end for**
11:    $\mathcal{V}^{meta'} \leftarrow \mathcal{V}^{meta'} \cup v^f$
12: **end for**
   {Sample a batch of original graph nodes and their $L$ hop neighbors}
13: $\mathcal{V}' \sim \mathcal{V}$
14: **for** $v \in \mathcal{V}'$ **do**
15:    $(\mathcal{V}^L, \mathcal{E}^L) \sim L\text{-HOP}(\mathcal{G} = (\mathcal{V}, \mathcal{E}), v)$
16:    $\mathcal{V}^{meta'} \leftarrow \mathcal{V}^{meta'} \cup \mathcal{V}^L$
17:    $\mathcal{E}^{meta'} \leftarrow \mathcal{E}^{meta'} \cup \mathcal{E}^L$
18: **end for**
   {Sample feature nodes and edges for sampled graph nodes to create a mini-batch for forward propagation}
19: **for** $v \in \mathcal{V}^{meta'}$ **do**
20:    **if** $v \in \mathcal{V}$ **then**
21:      $\mathcal{E}^{meta'} \leftarrow \mathcal{E}^{meta'} \cup \{(v, f) \mid f \in \mathcal{V}^f, \mathbf{X}[v, f] = 1\}$
22:    **end if**
23: **end for**
24: **Return** $\mathcal{G}^{meta'} = (\mathcal{V}^{meta'}, \mathcal{E}^{meta'})$

---

Table 5: Dataset statistics

| Dataset | # Nodes | # Edges | # Features | #Labels | $\mathbf{H}_{edge}$ |
|---|---|---|---|---|---|
| Cora | 2708 | 10556 | 1433 | 7 | 0.81 |
| CiteSeer | 3327 | 9104 | 3703 | 6 | 0.74 |
| Actor | 34493 | 495924 | 8415 | 5 | 0.22 |
| Chameleon | 7600 | 33544 | 931 | 5 | 0.23 |
| OGBN-Arxiv | 169343 | 1166243 | 128 | 40 | 0.81 |
| OGBN-Arxiv(year) | 169343 | 1166243 | 128 | 5 | 0.22 |
| Snap-Patents | 2923922 | 13975788 | 269 | 5 | 0.07 |

encodings for small-scale datasets, and node2vec based position encoding in the snap-patent dataset, as laplacian position encoding calculation is computationally infeasible at million-scale datasets, as proposed in GOAT, for large-scale datasets. These are further used in NAGPHORMER and LARGEGT. We select a number of negative features per node using hyperparameter tuning between the range of 5 to 30.

## C.4 CODEBASE

The codebase is available at https://anonymous.4open.science/r/nutag-7774/.

### C.5 LIMITATIONS AND FUTURE WORK

The major limitation of our work is that the proposed NEUTAG is not applicable to non-attributed graphs. Moreover, the proposed method is specifically designed for node classification tasks in both small and large-scale graphs. In contrast, graph classification tasks involve small graphs, e.g., pattern, cluster, zinc, peptides-func, peptides-struct (Dwivedi et al., 2023a; 2022b), having around 100 nodes on average. In such cases, dense GT has shown to perform exceptionally well without any computational challenges (Rampášek et al., 2022; Shirzad et al., 2023; Ying et al., 2021; Ma et al., 2023).

## D ADDITIONAL RESULTS

### D.1 ABLATION STUDY

Table 6: Ablation of NEUTAG on node classification task

| NEUTAG Variants | Cora | CiteSeer | Actor | Chameleon |
|---|---|---|---|---|
| NEUTAG (COMPLETE) | $87.26 \pm 2.14$ | $76.00 \pm 0.99$ | $\mathbf{36.25 \pm 2.43}$ | $\mathbf{65.26 \pm 2.43}$ |
| NEUTAG (LOCAL NBRS.) | $81.01 \pm 3.67$ | $75.49 \pm 1.10$ | $25.52 \pm 0.87$ | $30.70 \pm 1.49$ |
| NEUTAG$^+$ | $87.19 \pm 0.96$ | $\mathbf{77.68 \pm 1.9}$ | $34.93 \pm 0.83$ | $64.07 \pm 2.73$ |
| NEUTAG $-$F2F | $87.12 \pm 1.46$ | $75.70 \pm 0.3$ | $34.26 \pm 1.85$ | $63.02 \pm 3.79$ |
| NEUTAG$^+-$F2F | $\mathbf{87.67 \pm 1.10}$ | $74.65 \pm 0.95$ | $34.93 \pm 0.46$ | $64.12 \pm 1.95$ |

We design 5 variants of NEUTAG, **1)**NEUTAG (COMPLETE) which is the entire architecture **2)** NEUTAG (LOCAL NBRS.), which only consists of attention with local neighbors **3)**NEUTAG$^+$ consists of attentions with local neighbors, feature nodes and feature to feature attention path except attention with negative features **4)**NEUTAG-F2F consists of all attention paths except feature to feature attention and **5)**NEUTAG$^+$-F2F consists of local neighbor attention and feature node attention. Table 6 demonstrates the effectiveness of all the variants. Specifically, we observe that computing node representation by only attending to local neighbors NEUTAG (LOCAL NBRS) results in sub-optimal performance across all datasets. The performance drop is much more significant in non-homophilic graphs Actor and Chameleon. This signifies the crucial role of various attention paths involving feature nodes in learning both homophilic and heterophilic biases in NEUTAG. Table 6 also indicates that attention with local neighbors and feature nodes (NEUTAG$^+-$F2F) is competitive across all datasets. In contrast, additional attention with negative feature nodes and feature-to-feature attention NEUTAG (COMPLETE) provides a performance boost in heterophilic graph Actors and Chameleon.

### D.2 ADDITIONAL CHALLENGING HETEROPHILIC DATASETS

We further benchmark NEUTAG on challenging heterophilic datasets, with graph statistics and results summarized in Table 7. As shown, NEUTAG consistently outperforms scalable graph transformers by a significant margin. We further add

Table 7: Comparison of NEUTAG with scalable GT on additional challenging heterophilic graphs (Luan et al., 2024a)

| Dataset | # Nodes | # Edges | $\mathbf{H}_{edge}$ | NAGPHORMER | GOAT | LARGEGT | NEUTAG |
|---|---|---|---|---|---|---|---|
| Facebook | 4039 | 88234 | 0.5816 | $59.91 \pm 1.11$ | Error | Error | $\mathbf{63.09 \pm 1.29}$ |
| Cornell | 183 | 295 | 0.2983 | $58.90 \pm 5.23$ | $67.02 \pm 8.41$ | $53.51 \pm 11.89$ | $\mathbf{77.834 \pm 7.33}$ |
| Squirrel | 5201 | 217073 | 0.2234 | $38.05 \pm 2.00$ | $33.56 \pm 0.74$ | $36.5 \pm 2.69$ | $\mathbf{50.36 \pm 2.12}$ |
| Wisconsin | 251 | 499 | 0.1703 | $58.42 \pm 4.01$ | $74.11 \pm 7.76$ | $69.01 \pm 7.48$ | $\mathbf{78.034 \pm 4.19}$ |
| Texas | 183 | 309 | 0.0615 | $60.61 \pm 7.15$ | $68.64 \pm 4.09$ | $68.10 \pm 10.99$ | $\mathbf{82.69 \pm 4.04}$ |

### D.3 IMPACT OF MISSING FEATURES ON NEUTAG

We conduct two studies to examine how missing features affect NEUTAG. The first study looks at missing features only during the graph transformation stage. Here, the original input features are intact, but node–feature edges are randomly removed to simulate different levels of feature sparsity.

Table 8: Comparison of NEUTAG with scalable GT on additional NAGPHORMER datasets

| Dataset | # Nodes | # Edges | $\mathbf{H}_{edge}$ | NAGPHORMER | GOAT | LARGEGT | NEUTAG |
|---------|---------|---------|---------|-----------|------|---------|--------|
| Pokec | 1.6M | 30.62M | 0.4449 | $71.55 \pm 2.40$ | Error | $70.70 \pm 0.21$ | $\mathbf{71.97 \pm 0.22}$ |
| Photo | 7850 | 238163 | 0.8272 | $94.32 \pm 0.52$ | $\mathbf{95.58 \pm 0.45}$ | $93.76 \pm 0.89$ | $94.98 \pm 0.39$ |
| Computer | 13752 | 491722 | 0.7772 | $88.90 \pm 0.70$ | $\mathbf{91.55 \pm 0.59}$ | $87.39 \pm 1.10$ | $90.50 \pm 0.32$ |
| CoraFull | 19793 | 126842 | 0.5670 | $70.03 \pm 0.91$ | $69.21 \pm 0.64$ | $63.25 \pm 0.65$ | $\mathbf{72.62 \pm 0.44}$ |

We test this on the Chameleon dataset. Dropping feature connections reduces the number of feature–node edges, which lowers the homophily of the transformed graph $\mathcal{G}^{meta}$. This weakens the structural connectivity of the transformed graph and, as expected, leads to a drop in classification accuracy, as shown in the table below.

| **Feature Drop Rate in** $\mathcal{G}^{meta}$ | Homophily$_{ppr}^{\mathcal{G}}$ | Homophily$_{ppr}^{\mathcal{G}^{meta}}$ | **Accuracy** |
|---------|---------|---------|---------|
| 0% | 0.2349 | 0.2807 | $65.26 \pm 2.74$ |
| 50% | 0.2349 | 0.2615 | $60.51 \pm 2.15$ |
| 90% | 0.2349 | 0.2398 | $58.45 \pm 2.26$ |

Table 9: Effect of feature drop rate during graph transformation on homophily and accuracy.

In another study, for completeness, we now randomly drop features with varying probability ($p$) from the input node feature matrix itself and benchmark it against the baseline methods in table 10. We observe that NEUTAG maintains strong performance even under severe feature dropout, indicating robustness to missing input features. This is an encouraging result and aligns with our goal of designing scalable and generalizable graph transformers. That said, we believe a comprehensive study on robustness under various real-world noise and corruption settings is a significant task that warrants a separate investigation. In the present paper, our focus remains on proposing a scalable graph transformer architecture with strong theoretical foundations and empirical results.

# E  EXTENSION OF NEUTAG ON CONTINUOUS FEATURES

To convert real-valued features into binary form suitable for constructing feature nodes, we have employed a lightweight knowledge distillation framework. Specifically:

1. We first train a teacher model—a 2-layer MLP—using the original continuous node features for the node classification task.
2. Next, we train a student model that receives the same continuous features but passes them through a k-sparse encoder  (Makhzani & Frey, 2013) to produce intermediate binary encodings.
3. The student model is trained using a combination of: a) KL divergence loss between the soft predictions of the student and teacher models, and b) cross-entropy loss based on ground-truth node labels.

This approach enables us to retain task-relevant signal while producing binary features that can be used to define edges in the augmented graph for NEUTAG.

| Method | $p = 0$ | $p = 0.5$ | $p = 0.9$ |
|---|---|---|---|
| GRAPHSAGE | $48.95 \pm 3.16$ | $44.42 \pm 2.04$ | $37.50 \pm 2.81$ |
| GAT | $44.74 \pm 3.29$ | $38.33 \pm 3.73$ | $34.20 \pm 1.46$ |
| GIN | $32.68 \pm 3.68$ | $31.22 \pm 1.02$ | $30.26 \pm 3.54$ |
| MIXHOP | $47.68 \pm 2.89$ | $38.13 \pm 5.12$ | $31.14 \pm 3.54$ |
| LINKX | $48.20 \pm 3.31$ | $42.19 \pm 1.58$ | $35.26 \pm 3.04$ |
| GRAPHGPS | $42.88 \pm 1.88$ | $36.14 \pm 2.73$ | $32.96 \pm 3.77$ |
| EXPHORMER | $45.17 \pm 2.56$ | $42.45 \pm 1.60$ | $35.43 \pm 1.04$ |
| NAGPHORMER | $59.97 \pm 1.72$ | $56.72 \pm 1.90$ | $58.56 \pm 1.31$ |
| GOAT | $53.28 \pm 2.48$ | $43.85 \pm 1.93$ | $34.56 \pm 2.50$ |
| LARGEGT | $57.19 \pm 1.89$ | $55.24 \pm 2.45$ | $52.58 \pm 1.70$ |
| NEUTAG | $\mathbf{65.26 \pm 2.43}$ | $\mathbf{60.40 \pm 0.99}$ | $\mathbf{58.75 \pm 2.06}$ |

Table 10: Performance of different methods under varying feature drop rates ($p$).

