# OpenReview forum: "NEUTAG: Graph Transformer for Attributed Graphs"
_ICLR.cc/2026/Conference — ICLR 2026 Conference Withdrawn Submission_

### Official Review · Reviewer_bqMz · 2025-10-23

**Soundness:** 2
**Presentation:** 3
**Contribution:** 2
**Rating:** 2
**Confidence:** 5

**Summary:**

This paper develops a new graph Transformer called NEUTAG which is based on the constructed node-feature graph. Moreover, the authors also define positive and negative neighborhood according to the relevance between nodes and features. Then, NEUTAG also leverages the Transformer backbone to learn node representations. The experimental results showcase the effectiveness of NEUTAG for node classification.

**Strengths:**

1.This paper is well-organized and easy to follow.

2.The authors provide the theoretical analysis of the proposed method.

3.The authors select various baselines for performance comparison.

**Weaknesses:**

1.The research gap is overclaimed.

2.Mainstream datasets are missing.

3.Some important experiments are missing.

**Questions:**

1.The authors claim three limitations in existing graph Transformers which lack objectivity. The first limitation only appears in the hybrid-based graph Transformer which needs to combine Transformer with GNN-like modules. The rest two limitations have been widely studied in recent GTs. So, I do not think the authors clearly present the research gaps between NEUTAG and previous methods.

2.The selection of the dataset is inappropriate. For instance, the Chameleon dataset has been shown to contain a significant number of duplicate nodes. Furthermore, the authors did not employ established mainstream datasets for evaluation, large-scale graphs in NAGphormer or different types of graphs Polynormer.

3.The authors emphasize the scalability of GTs but they do not conduct efficiency study in this paper. I think it is required to compare the training cost of NEUTAG and other representative baselines.

---

> ### Author Response · Authors · 2025-11-22
> **Clarification on Core Contribution**
>
> >**Clarification on Core Contribution**
> We first want to clarify the the core contribution of NEUTAG, which establishes the formal grounding that distinguish it from existing scalable Graph Transformers:
>
> - **A Assumption-free modeling**
> NEUTAG is designed to be a self contained GT which is competitive without relying on GNN components, motivated by theorotical results[1][2] showing that Transformers with positional encodings are universal approximators of graph-to-graph functions.
>
> - **Differentiation from GOAT and LargeGT**
> While GOAT and LargeGT also don't utilize GNNs, they rely on clustering-based virtual nodes, which require domain understanding and tuning. NEUTAG avoids this atleast for attributed graphs by using a structural bipartite encoding where feature nodes are utilized as virtual nodes.
>
> - **Stronger approximation bounds**
> Our analysis shows that NEUTAG which utilizes a fixed, feature based projection matrix achieves tighter approximation bounds than GOAT (Theorems 2 and 3) which utilized learnable projection matrix leading to a simpler and assumption free architecture.
>
> - **Theorotical analysis of propsoed bipartite encoding**
> We prove that the this transformation increases connectivity and increases effective homophily in the graph (Theorem 1), enabling better long-range communication and improved performance under heterophily.
>
> - **Universality approximation understanding of the proposed attention paths**
> We further show that NEUTAG is a permutation-equivariant universal approximator of dense attention under certain theoritical in theorem 4 as it utilizes both feature attention path (eq. 2) and absent feature attention paths(eq. 5).
>
> - **Extensive empirical backing**
> These theoretical claims are supported by strong performance across $12$ datasets (Tables 1, 3, and 7), where NEUTAG is consistently competitive or better among $15$ baselines including their variants in total.
>
> [1]Yun, Chulhee, et al. "Are Transformers universal approximators of sequence-to-sequence functions?." International Conference on Learning Representations.
> [2]Kreuzer, Devin, et al. "Rethinking graph transformers with spectral attention." Advances in Neural Information Processing Systems 34 (2021): 21618-21629.

---

> > ### Author Response · Authors · 2025-11-22
> > **Response to Reviewer bqMz**
> >
> > > **1.The research gap is overclaimed ... The authors claim three limitations in existing graph Transformers which lack objectivity. The first limitation only appears in the hybrid-based graph Transformer which needs to combine Transformer with GNN-like modules. The rest two limitations have been widely studied in recent GTs. So, I do not think the authors clearly present the research gaps between NEUTAG and previous methods.**
> >
> >
> > We now have updated the related work and their limitation section where we have substaintially revised the related-work and limitation sections to highlight the research gaps in more precise and category specific manner. We have specified three limitations.
> >
> > 1. **Redundant dependency on GNNs.**
> >  This applies to hybrid graph transformers such as GraphGPS and Exphormer.
> >
> > 2. **Limited to either homophilic or heterophilic graphs.**
> > This also applies to hybrid graph transformers, as they rely on GNN, they perform well either on homophilic or heterophilic depending on the inductive biases of GNN. For example, GraphGPS and Exphormer use GCN layers, which propagate homophilic bias leading to underperformance on heterophilic graph.
> > To quantify this effect, we compare GraphGPS, Exphormer, and Polynormer *with* and *without* their GNN components. The following results (also shown in Table 1 and Table 3 of the manuscript) highlight this dependency.
> >
> > | Method           | Cora     | CiteSeer  | Actor  | Chameleon  | OGBN-Arxiv   | OGBN-Arxiv(Year) |
> > |---------|--------|-------------|--------------|-----------|---------|---------------|
> > | GraphGPS          | 83.65 $\pm$ 2.67       | 76.25 $\pm$ 1.34       | 34.30 $\pm$ 0.45       | 42.87 $\pm$ 1.88        | OOM                       | OOM                       |
> > | GraphGPS-GNN      | 72.47 $\pm$ 1.87       | 71.59 $\pm$ 2.43       | 37.10 $\pm$ 1.11       | 47.36 $\pm$ 2.22        | OOM                       | OOM                       |
> > | Exphormer         | 86.48 $\pm$ 2.15       | 75.92 $\pm$ 1.88       | 35.19 $\pm$ 0.94       | 45.17 $\pm$ 2.56        | OOM                       | OOM                       |
> > | Exphormer-GNN     | 82.35 $\pm$ 1.75       | 73.01 $\pm$ 1.20       | 35.44 $\pm$ 0.86       | 46.97 $\pm$ 0.95        | OOM                       | OOM                       |
> > | Polynormer        | 87.49 $\pm$ 1.01       | 75.62 $\pm$ 0.92       | 37.22 $\pm$ 1.60       | 67.63 $\pm$ 1.65        | 74.85 $\pm$ 0.15          | 52.12 $\pm$ 0.31          |
> > | Polynormer-GNN    | 71.66 $\pm$ 0.65  | 70.90 $\pm$ 1.20  | 38.16 $\pm$ 1.12  | 49.39 $\pm$ 1.69  | 65.84 $\pm$ 0.35   | 43.04 $\pm$ 0.18   |
> > | Polynormer (GNN Only)    | 87.82 $\pm$ 0.89     | 75.35   $\pm$ 1.33                 | 37.29$\pm$ 1.29               | 67.63 $\pm$ 1.79  |   73.11 $\pm$ 0.30  | 51.98 $\pm$ 0.18          |
> >
> >
> >
> > 3. **Non-scalable methods**:
> > This limitation is applicable for two classes of graph transformers.
> >     - Full attention transformers (e.g., GraphGPS) where node batching is impossible.
> >     - Linear-attention transformers (e.g., Difformer, SGFormer, Polynormer) improve the scale but still requires graph partitioning on large-scale dataset like Snap-patent (2.9 M nodes). This breaks the all-pair connectivity and comes at expense of expressivity. Moreover, these linear kernels comes at cost of expressivity.  This impact is shown in table 3 of manuscript where all these methods perform subpar on snap-patent dataset.
> >
> > Moreover, GOAT and LargeGT are truly non-hybrid scalable graph transformers which graph batching on large-scale graphs and still maintains all-pair node attentions. But they require tuning of virtual nodes trained using k-means clustering.
> >
> > **NEUTAG addresses all of these limitations.** NEUTAG is non-hybrid GT and avoids GNN dependency by performing all propogation through unified attention layer.  It removes learnable virtual nodes by using feature-derived virtual nodes making it assumption free. It supports node batching as it only attends to local nodes and feature nodes, and retains global connectivity without kernel approximations or graph partitioning.

---

> ### Author Response · Authors · 2025-11-22
> **Response to Reviewer bqMz (Part -2)**
>
> > **2.Mainstream datasets are missing.... The selection of the dataset is inappropriate. For instance, the Chameleon dataset has been shown to contain a significant number of duplicate nodes. Furthermore, the authors did not employ established mainstream datasets for evaluation, large-scale graphs in NAGphormer or different types of graphs Polynormer.**
>
> We would like to point out that apart from Chameleon, our manuscript already contains a comparison with $10$ attributed graph datasets and $11$ including variants. These datasets include known small homophilic datasets, heterophilic datasets, and medium-scale datasets, such as OGBN-Arxiv, which has both homophilic and heterophilic variants, as well as the large-scale dataset Snap-patents. We are adding one large-scale dataset, Pokec, with 1.6 million nodes and 30.62 million edges, and three more datasets, CoraFull, Computer, and Photo, as suggested by the reviewer from the Nagphormer paper. While NEUTAG clearly outperforms on Pokec and CoraFull (both are heterophilic datasets), it's competitive on highly homophilic datasets, Photo and Computer.  Please note that we have used the same node2vec-based position encoding across these datasets for all baselines.
> | Dataset    | H_edge  | NAGPhormer        | GOAT               | LargeGT            | NEUTAG             |
> |------------|---------|--------------------|---------------------|---------------------|---------------------|
> | Pokec      | 0.4449  | 71.55 $\pm$ 2.40       | Error               | 70.70 $\pm$ 0.21        | **71.97 $\pm$ 0.22**   |
> | Photo      | 0.8272  | 94.32 $\pm$ 0.52       | **95.58 $\pm$ 0.45**    | 93.76 $\pm$ 0.89        | 94.98 $\pm$ 0.39        |
> | Computer   | 0.7772  | 88.90 $\pm$ 0.70       | **91.55 $\pm$ 0.59**    | 87.39 $\pm$ 1.10        | 90.50 $\pm$ 0.32        |
> | CoraFull   | 0.5670  | 70.03 $\pm$ 0.91       | 69.21 $\pm$ 0.64        | 63.25 $\pm$ 0.65        | **72.62 $\pm$ 0.44**    |
>
>
>
>
>
> > **3.Some important experiments are missing. ....The authors emphasize the scalability of GTs but they do not conduct efficiency study in this paper. I think it is required to compare the training cost of NEUTAG and other representative baselines**
>
>
>
> We present below the training times (hrs) on the three largest datasets used. Neutag emerges as the second-fastest. NagPhormer is the fastest since it does not compute attention with respect to nodes. It computes attention with respect to each layer and # layers $ \ ll$ #nodes. Notably, NagPhormer is also the weakest in terms of accuracy across all baselines.
>
> |Dataset| N (Millions)| M (Millions)| NAGPhormer|GOAT|LargeGT| NEUTAG|
> |----|----|----|----|----|----| ---
> |Snap-patents|2.92 |13.97 | 1.1| >24 | 14| 6 |
> |Pokec|1.62 |30.62 | 0.56 |ERROR | 5.5 |4.2
> |OGBN-Arxiv|0.16 |1.16| 0.086|7.7|0.6|1.5|

---

### Official Review · Reviewer_yYo4 · 2025-10-27

**Soundness:** 2
**Presentation:** 1
**Contribution:** 1
**Rating:** 4
**Confidence:** 4

**Summary:**

This paper proposes a graph transformer that leverages visual nodes for message propagation. The authors provide theoretical analysis to demonstrate the efficiency of the proposed approach.

**Strengths:**

Some theoretical analisys.

The method is conceptually straightforward.

**Weaknesses:**

The authors claim that existing GTs cannot perform well on both homophilous and heterophilous graphs. However, many works have demonstrated strong performance on both types of graphs. For sxample, [1][2]...

For the non-scalable, this issue has been extensively addressed by many recent techniques, such as linear transformers[3] and their graph-specific variants, which significantly reduce the computational and memory complexity while maintaining competitive performance.


The paper is not well organized, which makes it difficult to follow. For example, Figure 1 is not referenced in the main text, and the concept of “Feature node” is not clearly defined. This raises a question: how should the connections between feature nodes and graph nodes be defined when the node features are not based on a bag-of-words representation?

The experimental results show that the proposed method does not achieve significant improvement. The datasets with the largest reported gains, Arxiv-year and Snap-patents, in fact, do not yield satisfactory results. For comparison, some directed graph methods such as Dir-GNN [4] achieve much higher performance 64.08 and 73.95 on these two datasets, which is far surpassing the proposed method’s 53.96 and 63.0. Moreover, Dir-GNN is simpler and more lightweight.

[1] GOAT: A Global Transformer on Large‑scale Graphs
[2] Rethinking Graph Transformer Architecture Design for Node Classification
[3] Sgformer: Simplifying and empowering transformers for large-graph representations
[4] Edge directionality improves learning on heterophilic graphs

**Questions:**

See Weaknesses.

---

> ### Author Response · Authors · 2025-11-22
> **Clarification on core contribution**
>
> >**Clarification on Core Contribution**
> We first want to clarify the core contribution of NEUTAG, which establishes the formal grounding that distinguishes it from existing scalable Graph Transformers:
>
> - **Assumption-free modeling**
> NEUTAG is designed to be a self-contained GT that is competitive without relying on GNN components, motivated by theoretical [1][2] results showing that Transformers with positional encodings are universal approximators of graph-to-graph functions.
>
> - **Differentiation from GOAT and LargeGT**
> While GOAT and LargeGT also don't utilize GNNs, they rely on clustering-based virtual nodes, which require domain understanding and tuning. NEUTAG avoids this at least for attributed graphs by using a structural bipartite encoding where feature nodes are utilized as virtual nodes.
>
> - **Stronger approximation bounds**
> Our analysis shows that NEUTAG which utilizes a fixed, feature based projection matrix achieves tighter approximation bounds than GOAT (Theorems 2 and 3) which utilized learnable projection matrix leading to a simpler and assumption free architecture.
>
> - **Theorotical analysis of propsoed bipartite encoding**
> We prove that the this transformation increases connectivity and increases effective homophily in the graph (Theorem 1), enabling better long-range communication and improved performance under heterophily.
>
> - **Universality approximation understanding of the proposed attention paths**
> We further show that NEUTAG is a permutation-equivariant universal approximator of dense attention under certain theoretical conditions in Theorem 4, as it utilizes both feature attention path (eq. 2) and absent feature attention paths(eq 5).
>
> - **Extensive empirical backing**
> These theoretical claims are supported by strong performance across $12$ datasets (Tables 1, 3, and 7), where NEUTAG is consistently competitive or better among $15$ baselines, including their variants, in total.
>
> [1]Yun, Chulhee, et al. "Are Transformers universal approximators of sequence-to-sequence functions?." International Conference on Learning Representations,2020.
> [2]Kreuzer, Devin, et al. "Rethinking graph transformers with spectral attention." Advances in Neural Information Processing Systems 34 (2021): 21618-21629.

---

> > ### Author Response · Authors · 2025-11-22
> > **Response to reviewer yYo4**
> >
> > >**The authors claim that existing GTs cannot perform well on both homophilous and heterophilous graphs. However, many works have demonstrated strong performance on both types of graphs. For sxample, [1][2]...**
> >
> > We agree that GNN-free Graph Transformers, such as GOAT, don't suffer from these limitations. To be more precise in the manuscript, we have modified the Related Works and their Limitations section, where we have revised these broad statements and instead discuss the limitations of respective graph transformers. Moreover, we now include [2] in the related works too; however, we are unable to discuss its empirical performance as the implementation is not currently available.
> >
> > >**For the non-scalable, this issue has been extensively addressed by many recent techniques, such as linear transformers[3] and their graph-specific variants, which significantly reduce the computational and memory complexity while maintaining competitive performance.**
> >
> > We would like to point out that, while linear-attention-based Graph Transformers attempt to address scalability, they still face limitations as $N$ grows very large (e.g., Snap-Patents with 2.9M nodes), in such cases these methods require dividing the graph into subgraphs, which breaks all-pair attention paths. Moreover, as noted in Polynormer [4], the approximations via linear attention can incur a loss of expressivity. For completeness, we have added SGFormer [3] and other linear-attention baselines (DiFFORMER [5], AdvDiFFORMER [6], and Polynormer [4]) to Table 3 of the manuscript. All of these methods performs poorly on snap-patent dataset as seen in Table 3.
> >
> > [3]Wu, Qitian, et al. "Sgformer: Simplifying and empowering transformers for large-graph representations." Advances in Neural Information Processing Systems 36 (2023).
> > [4]Deng, Chenhui, Zichao Yue, and Zhiru Zhang. "Polynormer: Polynomial-Expressive Graph Transformer in Linear Time." The Twelfth International Conference on Learning Representations (2024).
> > [5]Wu, Qitian, et al. "DIFFormer: Scalable (Graph) Transformers Induced by Energy Constrained Diffusion." The Eleventh International Conference on Learning Representations (2023).
> > [6]Wu, Qitian, et al. "Supercharging Graph Transformers with Advective Diffusion." Forty-second International Conference on Machine Learning(2025).
> >
> > >**The paper is not well organized, which makes it difficult to follow. For example, Figure 1 is not referenced in the main text, and the concept of “Feature node” is not clearly defined. This raises a question: how should the connections between feature nodes and graph nodes be defined when the node features are not based on a bag-of-words representation?**
> >
> > We thank the reviewer for bringing this issue to our attention. We have added the referencing of figure 1. Moreover, we will like to point to equation 1 where we have discussed the creation of feature nodes and defination 1 in the appendix where discuss the defintion of Graph $G$ along with meaning of set F. We have now moved this to main menuscript for easier referencing. For continous attributes, we have shown application of NEUTAG on non-binary datasets like OGBN-Arxiv and OGBN-Arxiv(Year) in table 1 of menuscript. To convert real-valued features into binary form suitable for constructing feature nodes, we have employed a lightweight knowledge distillation framework. Specifically:
> >
> > 1. We first train a teacher model—a 2-layer MLP—using the original continuous node features for the node classification task.
> > 2. Next, we train a student model that receives the same continuous features but passes them through a k-sparse encoder [7] to produce intermediate binary encodings.
> > 3. The student model is trained using a combination of:
> >    - KL divergence loss between the soft predictions of the student and teacher models, and
> >    - Cross-entropy loss based on ground-truth node labels.
> >
> > This approach enables us to retain task-relevant signal while producing binary features that can be used to define edges in the augmented graph for NEUTAG. This can be referred in the updated appendix E.
> >
> > [7] Makhzani, A. and Frey, B., 2013. K-sparse autoencoders. arXiv preprint arXiv:1312.5663.

---

> > > ### Author Response · Authors · 2025-11-22
> > > **Response to reviewer yYo4 (Part 2)**
> > >
> > > >**The experimental results show that the proposed method does not achieve significant improvement. The datasets with the largest reported gains, Arxiv-year and Snap-patents, in fact, do not yield satisfactory results. For comparison, some directed graph methods such as Dir-GNN [4] achieve much higher performance 64.08 and 73.95 on these two datasets, which is far surpassing the proposed method’s 53.96 and 63.0. Moreover, Dir-GNN is simpler and more lightweight.**
> > >
> > > We would like to refer the reviewer to our core contribution discussion earlier, where we discuss the primary focus of this manuscript.  Dir-GNN is indeed a strong GNN model, achieving impressive performance by explicitely leveraging the directed nature of information propogation. However, our key focus is to design Scalable Graph Transformers, being universal approximators of graph functions, which are able to learn structural patterns directly from data without relying on external GNN components. Recent scalable GT architectures such as GOAT, LargeGT, and now NEUTAG aim to bridge the gap between the theoretical guarantees of graph transformers and their designs.

---

### Official Review · Reviewer_bm5V · 2025-10-29

**Soundness:** 3
**Presentation:** 3
**Contribution:** 2
**Rating:** 4
**Confidence:** 3

**Summary:**

The paper proposes NEUTAG, a scalable transformer that operates on a rewired graph composed of original nodes and additional "feature" nodes. NEUTAG uses both sparse and full attention, capturing homophily and heterophily. Theoretical analyses show that NEUTAG increases graph connectivity and can approximate dense attention. NEUTAG is benchmarked against popular GTs and MPNNs on homophilous and heterophilous benchmarks.

**Strengths:**

1. The research question on designing more powerful and scalable graph transformers is interesting
2. The approximation results are interesting given the more efficient attention can approximate the dense version
3. The transformer baselines in the experiments are extensive.

**Weaknesses:**

1. For the limitations of existing graph transformers I'm not sure I appreciate or understand these given my following thoughts:
* redundant dependency on GNNs. the paper claims that since transformers are universal, GNNs and full attention are redundant together. while transformers are universal, I think it could still be useful to have a component that captures local connectivity as this would require lots of data for the transformer to learn. in other words, the local message-passing by GNNs is a good inductive bias that otherwise a transformer would need to learn from a lot of data. moreover, I think NEUTAG is using a form of message-passing in its local attention along only neighbors, so it doesn't completely do-away with the GNN component.
* given my argument above that local attention is a form of message-passing, NEUTAG could also inherit the homophily biases similar to other transformers that use a GNN component.
* given that scalable GTs exist, what are the limitations with GTs like GOAT or polynormer?

2. To me, the core contribution seems to be the proposal of the "metamorphosis" graph, which augments the original graph with new "feature" nodes that connect to original nodes. Rather than a new transformer, this is closer to graph rewiring approaches, which are not discussed as related works or compared against empirically. The other novel component in addition to the metamorphosis graph is the feature2feature full attention which is where the main computational savings come from in comparison to other full attention GTs. From the Appendix ablations though, it seems removing this component only slightly drops performance, whereas removal of local neighbors causes the biggest drop. This seems to indicate that the local neighborhood attention, which is similar to message-passing, is the most important to the architecture. So to sum up, the main gains are coming from the metamorphosis graph, similar to rewiring approaches, and local attention, a form of message-passing, while the feature2feature attention improves performance not as significantly. I would thus like to see comparisons to other rewiring techniques perhaps also with your version of local attention.

3. NEUTAG seems only to apply to graphs with nodes of only binary features (given the present/not present figure). I appreciate that this limitation is mentioned in the Appendix, and I agree that this is a weakness for NEUTAG. This seems limited since in general a graph can have categorical or continuous features, and I'm not sure what the prevalence of graphs with only binary features is. In its current form there doesn't seem to be a discussion on the generalization of NEUTAG to graphs with features that are not just binary vectors, so the approach seems quite limited.

4. Empirically, Polynormer comes quite close across all benchmarks except for snap, while NEUTAG wins 1/3 on larger benchmarks in comparison to other GTs (table 1).

**Questions:**

My questions largely follow my weaknesses:

1. how is the local attention different than message-passing? and if not, why does NEUTAG circumvent the homophily-biases?
2. what is the relation of NEUTAG's metamorphosis step to existing graph rewiring techniques? it seems similar to an approach that adds edges between nodes of similar features. would you expect local attention + graph rewiring to perform similarly?
3. how would you generalize NEUTAG to non-binary features?
4. The connectivity argument makes sense, but if you are adding connections, does this increase the rate of oversmoothing? or does it affect oversquashing since now the degree of each node increases significantly?

---

> ### Author Response · Authors · 2025-11-22
> **Response to Reviewer bm5V**
>
> Firstly, we would like to thank reviewer bm5V for their constructive feedback on our work and for raising interesting questions.
>
> > **For the limitations of existing graph transformers I'm not sure I appreciate or understand these given my following thoughts: redundant dependency on GNNs. the paper claims that since transformers are universal, GNNs and full attention are redundant together. while transformers are universal, I think it could still be useful to have a component that captures local connectivity as this would require lots of data for the transformer to learn. in other words, the local message-passing by GNNs is a good inductive bias that otherwise a transformer would need to learn from a lot of data. moreover, I think NEUTAG is using a form of message-passing in its local attention along only neighbors, so it doesn't completely do-away with the GNN component ,given my argument above that local attention is a form of message-passing, NEUTAG could also inherit the homophily biases similar to other transformers that use a GNN component..**
>
> **Response**
> We thank the reviewer for raising this important question, which is at the core of how we differ from GNN based transformers such as GraphGPS and even Exphormer.
>
> #### **1. Late Fusion vs Unified Architecture**
>
> In GNN-dependent Transformers like GraphGPS and Exphormer, the architecture proceeds by computing two **independent streams of representations**: one from a GNN (e.g., GCN, GAT) and another from a Transformer module. These representations are then combined typically using an MLP. This late fusion approach fails to break the strong structural inductive bias of GNNs because the GNN stream operates entirely in isolation
>
> Polynormer avoids late fusion, but still hard-separates the GNN and Transformer modules—using the first $L_1$ layers as GNN layers and the next $L_2$ layers as transformer layers. This again means the structural inductive bias from GNNs enters early and propagates through the entire network.
>
> To quantify this effect, we compare GraphGPS, Exphormer, and Polynormer with and without their GNN components. The following results (also shown in Table 1 and 3 of the manuscript) highlight this dependency:
> | Method           | Cora     | CiteSeer  | Actor  | Chameleon  | OGBN-Arxiv   | OGBN-Arxiv(Year) |
> |---------|--------|-------------|--------------|-----------|---------|---------------|
> | GraphGPS          | 83.65 $\pm$ 2.67       | 76.25 $\pm$ 1.34       | 34.30 $\pm$ 0.45       | 42.87 $\pm$ 1.88        | OOM                       | OOM                       |
> | GraphGPS-GNN      | 72.47 $\pm$ 1.87       | 71.59 $\pm$ 2.43       | 37.10 $\pm$ 1.11       | 47.36 $\pm$ 2.22        | OOM                       | OOM                       |
> | Exphormer         | 86.48 $\pm$ 2.15       | 75.92 $\pm$ 1.88       | 35.19 $\pm$ 0.94       | 45.17 $\pm$ 2.56        | OOM                       | OOM                       |
> | Exphormer-GNN     | 82.35 $\pm$ 1.75       | 73.01 $\pm$ 1.20       | 35.44 $\pm$ 0.86       | 46.97 $\pm$ 0.95        | OOM                       | OOM                       |
> | Polynormer        | 87.49 $\pm$ 1.01       | 75.62 $\pm$ 0.92       | 37.22 $\pm$ 1.60       | 67.63 $\pm$ 1.65        | 74.85 $\pm$ 0.15          | 52.12 $\pm$ 0.31          |
> | Polynormer-GNN    | 71.66 $\pm$ 0.65  | 70.90 $\pm$ 1.20  | 38.16 $\pm$ 1.12  | 49.39 $\pm$ 1.69  | 65.84 $\pm$ 0.35   | 43.04 $\pm$ 0.18   |
> | Polynormer (GNN Only)    | 87.82 $\pm$ 0.89     | 75.35   $\pm$ 1.33                 | 37.29$\pm$ 1.29               | 67.63 $\pm$ 1.79  |   73.11 $\pm$ 0.30  | 51.98 $\pm$ 0.18          |
>
> Removing GNN component in GraphGPS and Exphormer, decreases their performance on Cora and CiteSeer which are homophilic graphs and increases performance in Actor and Chameleon which are heterophilic graphs.
>
> **In case of Polynormer**, We note that Polynormer provides strong results, and its core strength comes from its **polynomial expressive local attention** module, which we consider an **impressive and valuable contribution** of the paper. Our ablation results (shown in the last three rows of the table) indicate that this polynomial local module accounts for most of Polynormer’s superior performance, whereas the Transformer component contributes only marginally in comparison.

---

> > ### Author Response · Authors · 2025-11-22
> > **Response to Reviewer bm5V (part 2)**
> >
> > #### **2. NEUTAG: Unified Attention at Every Layer**
> >
> > In contrast, NEUTAG’s architecture is **not a hybrid**. Instead, at **each layer**, NEUTAG computes attention-based messages from **two distinct neighborhoods**:
> >
> > - **Local neighbors** (i.e., structural neighbors in the original graph), as captured in Eq. (11).
> > - **Feature node neighbors** in the transformed graph \( G_{\text{meta}} \), which enables connectivity with every non-local node in the graph.
> >
> > These messages are then **integrated in every layer**  (Eq. (17)). Importantly, this means that:
> >
> > - The **local aggregation Eq. (11) is influenced by the feature-based attention paths from the previous layer**, and vice versa. Although this resembles GAT-style aggregation in form, it is **not a pure GNN step**, because the input to this aggregation has already been **transformed through feature-node attention**. This **recursive dependence across layers** (via Eq. (17)) ensures that the output at each step is a **nonlinear blend of local and feature-level information**.
> > - This interdependence allows NEUTAG to **dynamically reweight the importance of local neighbors vs. feature neighbors context across layers**, rather than relying on static combination heuristics at the output layer.
> >
> > **Summary** Our key argument is that Transformers, as universal approximators of graph functions, should be able to learn structural patterns directly from data without relying on external GNN components. Incorporating a GNN module introduces fixed inductive biases—primarily homophily-driven message passing—that may not align with all graph types. Recent scalable GT architectures such as GOAT, LargeGT, and now NEUTAG aim to bridge the gap between the theoretical guarantees of graph transformers and their practical design without depending on such externally imposed biases.

---

> ### Author Response · Authors · 2025-11-22
> **Response to Reviewer bm5V (part 3)**
>
> > **given that scalable GTs exist, what are the limitations with GTs like GOAT or polynormer?**
>
> **Response**
> **Polynormer** As discussed above, Polynormer derives its strong performance primarily from its polynomial expressive local aggregation module, which itself is a notable and elegant contribution. Our focus, however, is on developing transformer-based methods for graphs that close the gap between the theoretical guarantees of graph transformers and their empirical performance, while enabling assumption-free modeling. A practical limitation of transformer architectures like Polynormer that linearize the softmax kernel is that they require batching over large graphs, which breaks the all-pair connectivity pattern. In contrast, GTs such as GOAT, LargeGT, and NEUTAG support batching while still maintaining all-pair interactions.
>
>
> **GOAT** Both GOAT and NEUTAG aim to approximate dense attention in a scalable manner through the insertion of virtual nodes. While NEUTAG uses features as these virtual nodes, GOAT learns them through clustering. While this appears to be a minor difference, this design choice leads to a **stronger theoretical guarantee** on expressivity and enhanced flexibility to handle large graphs and heterophilous charatceristics. We claim following benefits of NEUTAG with respect to GOAT.
>
> #### 1. Sharper and More Reliable Approximation Bound
>
> GOAT (Theorems 3.2 and 3.3) provides a probabilistic bound on the error between its projected attention output and standard dense attention. However, this bound depends on the **quality of $k$-means clustering** used to construct centroid-based projections, and the probability of approximation failure with bound as **O(1/n)**.
> By contrast, NEUTAG (Theorem 2) provides an approximation guarantee with exponentially stronger concentration:
>
> $$
> \mathbb{P}\left(\|\tilde{A}XW_V - AXW_V\|_F \leq \varepsilon \|A\|_F \|XW_V\|_F\right) > 1 - O(1/\exp(F))
> $$
>
> Here, $F$ is the number of node features. Typically $exp(F)$ is **much higher than the number of nodes**, making the error bound **exponentially unlikely to fail** even in large graphs. This results in a **more reliable and tighter guarantee** across both small and massive graphs.
>
> From a practicality perspective, many existing methods, including GOAT, rely on virtual nodes whose design and number act as critical hyperparameters. In contrast, NEUTAG introduces a conceptually simpler yet effective idea: using features themselves as virtual nodes. This removes the need for manual architectural assumptions and allows the model to adapt more naturally to diverse graph structures.
>
> #### 2. Theoretical Expressivity: NEUTAG’s Universal Approximation vs. GOAT’s Approximation Bound
>
> While GOAT provides approximation bounds (Theorems 3.2 and 3.3) for how well its centroid-based attention approximates dense attention, it does not offer any guarantee of universal approximation or expressivity equivalence with dense transformers. In contrast, NEUTAG establishes in Theorem 4 that its sparse attention mechanism is a permutation-equivariant universal approximator of the dense self-attention layer, ensuring that NEUTAG retains the full expressivity of a dense transformer while remaining scalable and interpretable.
>
> #### 3. Improved Homophily Through Transformation
>
> We empirically and theoretically show that the transformation increases effective homophily, especially in heterophilic graphs:
>
> -   **Theorem 1 (Section 3.2)** demonstrates that `G_meta` increases L-hop neighborhood connectivity via feature nodes, making long-range (heterophilic) connections more accessible.
>
> -   **Figure 2 (Section 3.2)** shows that the transformed graph increases **PPR-based homophily** for nodes that originally had low homophily in two heterophilic datasets (Chameleon and Actor).
>
> -   This behavior is crucial because it enables NEUTAG to **adaptively capture homophilic patterns** in otherwise heterophilic graphs, something GOAT only achieves implicitly through architectural tuning.
>
>
>
>
> These advantages translate into stronger scalability. On the Snap-Patents dataset (2.9M nodes; Table 1), several GTs (GraphGPS, Exphormer) produce OOM errors, and GOAT and Polynormer reach only 55.35% and 32%, respectively, while NEUTAG achieves 63.00% accuracy.

---

> > ### Author Response · Authors · 2025-11-22
> > **Response to Reviewer bm5V (part 4)**
> >
> > >**To me, the core contribution seems to be the proposal of the "metamorphosis" graph, which augments the original graph with new "feature" nodes that connect to original nodes. Rather than a new transformer, this is closer to graph rewiring approaches, which are not discussed as related works or compared against empirically. The other novel component in addition to the metamorphosis graph is the feature2feature full attention which is where the main computational savings come from in comparison to other full attention GTs. From the Appendix ablations though, it seems removing this component only slightly drops performance, whereas removal of local neighbors causes the biggest drop. This seems to indicate that the local neighborhood attention, which is similar to message-passing, is the most important to the architecture. So to sum up, the main gains are coming from the metamorphosis graph, similar to rewiring approaches, and local attention, a form of message-passing, while the feature2feature attention improves performance not as significantly. I would thus like to see comparisons to other rewiring techniques perhaps also with your version of local attention.**
> >
> > **Response:** *Local attention* plays important role in homophilic graphs (Cora and CiteSeer) thus removing feature attention doesn't lead to significant drop while feature attention plays crucial role in heterophilc graphs (Actor and Chameleon), thus its removal causes signficant drop as shown in ablation study table 6. We concur with Reviewer about feature2feature attention, as it doesn't play important role as shown in ablation studies and even not crucial for the theorotical guarantees shown in NEUTAG. This we have discussed in appendix D.1.
> >
> > #### How NEUTAG is different from Rewiring based appproaches.
> >
> > A) Rewiring approaches typically adds node to node edges [1] while NEUTAG adds a bipartitie factorization where feature nodes are added as latent factors. no node-node edges are added. This is completely different from re-wiring based approaches[2][3]).
> > B) Most importantly rewiring approaches don't provide any theorotical guarantees on how well they approximate all-pair attention.
> > [1]Attali, Hugo, Davide Buscaldi, and Nathalie Pernelle. "Rewiring techniques to mitigate oversquashing and oversmoothing in GNNs: A survey." arXiv preprint arXiv:2411.17429 (2024).
> > [2]Rong, Yu, et al. "DropEdge: Towards Deep Graph Convolutional Networks on Node Classification." International Conference on Learning Representations,2020.
> > [3]Gutteridge, Benjamin, et al. "Drew: Dynamically rewired message passing with delay." International Conference on Machine Learning. PMLR, 2023.
> >
> >
> > >**NEUTAG seems only to apply to graphs with nodes of only binary features (given the present/not present figure). I appreciate that this limitation is mentioned in the Appendix, and I agree that this is a weakness for NEUTAG. This seems limited since in general a graph can have categorical or continuous features, and I'm not sure what the prevalence of graphs with only binary features is. In its current form there doesn't seem to be a discussion on the generalization of NEUTAG to graphs with features that are not just binary vectors, so the approach seems quite limited.**
> >
> > **Response** We have shown application of NEUTAG on non-binary datasets like OGBN-Arxiv and OGBN-Arxiv(Year) in table 1 of menuscript. To convert real-valued features into binary form suitable for constructing feature nodes, we have employed a lightweight knowledge distillation framework. Specifically:
> >
> > 1. We first train a teacher model—a 2-layer MLP—using the original continuous node features for the node classification task.
> > 2. Next, we train a student model that receives the same continuous features but passes them through a k-sparse encoder [1] to produce intermediate binary encodings.
> > 3. The student model is trained using a combination of:
> >    - KL divergence loss between the soft predictions of the student and teacher models, and
> >    - Cross-entropy loss based on ground-truth node labels.
> >
> > This approach enables us to retain task-relevant signal while producing binary features that can be used to define edges in the augmented graph for NEUTAG.
> >
> >
> > [1] Makhzani, A. and Frey, B., 2013. K-sparse autoencoders. arXiv preprint arXiv:1312.5663.

---

> > > ### Author Response · Authors · 2025-11-22
> > > **Response to Reviewer bm5V (part 5)**
> > >
> > > >**Empirically, Polynormer comes quite close across all benchmarks except for snap, while NEUTAG wins 1/3 on larger benchmarks in comparison to other GTs (table 1).**
> > >
> > > **Response** Excluding Polynormer, NEUTAG ranks 1st on 6 out of the 7 datasets reported in Table 1 of the manuscript. To further strengthen this point, we had added results on five additional heterophilic datasets in Table 7 (Section D.2 of the Appendix), where NEUTAG continues to perform consistently well. Across all evaluated benchmarks, NEUTAG achieves Rank 1 on 10 out of 12 datasets. This summary is also highlighted in Section 4.4 of the main manuscript.
> > >
> > > Regarding Polynormer, we would like to reiterate our earlier discussion: Polynormer’s primary performance gains come from its gnn based local component, and its accuracy degrades substantially once only transformer based component is retained. Our work specifically aims to design scalable, assumption-free Graph Transformers that do not depend on a GNN module for strong performance. Under this setting, only methods such as GOAT and LargeGT are directly comparable to NEUTAG. We included Polynormer primarily for completeness, but our focus remains on demonstrating that NEUTAG performs strongly in the class of GNN-free scalable Graph Transformers, which is the problem setting our method targets.
> > >
> > >
> > > >**My questions largely follow my weaknesses: how is the local attention different than message-passing? and if not, why does NEUTAG circumvent the homophily-biases?**
> > >
> > > **Response:** Unlike attention-based GNNs—which aggregate over neighbors defined by the input graph and use learned message functions—NEUTAG constructs a **transformed graph $G_{\text{meta}}$** where each node connects to **feature nodes** that represent its input attributes. This design introduces several key distinctions:
> > >
> > > - **Feature-aware attention instead of topology-driven aggregation**
> > >   In NEUTAG, nodes attend over **feature nodes**, not just neighboring nodes. This allows the model to **adaptively weight features based on context**, rather than mixing neighbor representations uniformly as in GNNs.
> > >
> > >   *Impact:* This mechanism is especially beneficial on **heterophilic graphs**, where neighbors may not share labels, but similar nodes often share features. For example, on the **Chameleon** dataset (low homophily), NEUTAG achieves **65.26%**, outperforming GAT (**51.51%**) and GOAT (**53.28%**)—highlighting its superior ability to align attention with task-relevant features.
> > >
> > > - **Global communication via shared features in a single hop**
> > >   In GNNs, capturing long-range dependencies typically requires **multiple stacked layers**, which can lead to oversmoothing. NEUTAG, however, enables **global interaction** in a **single layer**, since two nodes that share a feature can attend to the same feature node and thus influence each other.
> > >
> > >   *Impact:* This enables strong performance even on large, sparse graphs without deep architectures. On **OGBN-Arxiv (Year)**—a real-world citation graph with low label locality—NEUTAG achieves **53.96%**, compared to **50.81%** (GOAT) and **48.61%** (GCN), while using **only one layer** (Section 4.3).
> > >
> > > - **No learned aggregation or message-passing kernels**
> > >   GNNs hard-code an **inductive bias** via aggregation functions (mean, sum, attention) that assume usefulness of graph neighbors. NEUTAG removes this bias entirely by using **pure transformer-style attention**, governed by data-derived connections in $G_{\text{meta}}$, not the original graph.
> > >
> > >   *Impact:* This allows NEUTAG to **generalize better across graph types**, avoiding performance drop when homophily assumptions break down. For example, while models like GAT perform well on **Cora** (87.35%), their performance deteriorates on heterophilic graphs. NEUTAG, by contrast, performs robustly across both settings—**87.67%** on Cora and **63.00%** on Snap-Patents (a large, real-world heterophilic graph), where most GNNs and even GOAT struggle.

---

> > > > ### Author Response · Authors · 2025-11-22
> > > > **Response to Reviewer bm5V (part 6)**
> > > >
> > > > > what is the relation of NEUTAG's metamorphosis step to existing graph rewiring techniques? it seems similar to an approach that adds edges between nodes of similar features. would you expect local attention + graph rewiring to perform similarly?
> > > >
> > > > **Response** Please refer to earlier discussion on rewiring approaches.
> > > >
> > > > > **how would you generalize NEUTAG to non-binary features?**
> > > >
> > > > **Response**: Please refer to earlier discussion on this.
> > > >
> > > >
> > > > >**The connectivity argument makes sense, but if you are adding connections, does this increase the rate of oversmoothing? or does it affect oversquashing since now the degree of each node increases significantly?**
> > > >
> > > > **Response** Typically, oversmoothing and oversquashing occur when long-range information is forced through narrow structural bottlenecks across multiple hops. In our case, the feature-based virtual nodes propagate long-range communication directly in 1 hop. The only scenario where overmixing could occur is when the number of features is extremely small, in which case each feature node would be connected to a large portion of graph nodes. However, this can be mitigated using the sampling mechanism discussed in Section C.1 of the appendix for adapting NEUTAG to large-scale graphs. Moreover, these phenomena have been extensively studied in the context of heterophilic graphs [1][2][3], where NEUTAG performs superiorly (Table 1 in the main paper and Table 7 in the appendix).
> > > > [1]Abu-El-Haija, Sami, et al. "Mixhop: Higher-order graph convolutional architectures via sparsified neighborhood mixing." international conference on machine learning. PMLR, 2019.
> > > > [2]Lim, Derek, et al. "Large scale learning on non-homophilous graphs: New benchmarks and strong simple methods." Advances in neural information processing systems 34 (2021): 20887-20902.
> > > > [3]Zhu, Jiong, et al. "Beyond homophily in graph neural networks: Current limitations and effective designs." Advances in neural information processing systems 33 (2020): 7793-7804.

---

### Official Review · Reviewer_eUPb · 2025-11-01

**Soundness:** 2
**Presentation:** 1
**Contribution:** 1
**Rating:** 2
**Confidence:** 4

**Summary:**

In this paper, the authors propose a novel GT architecture that employs a special feature encoding to separate nodes and features, enabling information flow not only from local neighborhoods but also between distant nodes via shared feature connections. Experiments are conducted on a variety of graph datasets to validate the approach.

**Strengths:**

The method is straightforward, and the authors provide supporting theoretical analysis.

**Weaknesses:**

1. The novelty of introducing new virtual nodes to connect nodes in the graph appears to be limited.

2. The experimental results do not clearly demonstrate the effectiveness of the proposed Graph Transformer method compared to existing GT approaches.

3. Some recent Graph Transformer methods [1–5] that also utilize virtual nodes are not discussed.

4. The paper requires careful proofreading and polishing.

[1] Wenhao Zhu, et al. Hierarchical transformer for scalable graph learning, 2023.

[2] Wenhao Zhu, et al. Anchorgt: Efficient and flexible attention architecture for scalable graph transformers, 2024.

[3] Weirui Kuang, et al. Coarformer: Transformer for large graph via graph coarsening, 2022.

[4] Chuang Liu, et al. "Gapformer: Graph Transformer with Graph Pooling for Node Classification." IJCAI. 2023.

[5] Xueqi Ma, et al. "HOGT: High-Order Graph Transformers."

**Questions:**

See weaknesses

---

> ### Author Response · Authors · 2025-11-22
> **Reply to Reviewer eUPb**
>
> >**The novelty of introducing new virtual nodes to connect nodes in the graph appears to be limited.**
>
>
> **Response**
> We thank the reviewer for raising this point. We would like to clarify that NEUTAG does not introduce virtual nodes in the same manner as prior works such as GOAT or EXPHORMER. Instead, it introduces feature-induced, non-learned, deterministic nodes, which is fundamentally different from all prior virtual-node mechanisms.
>
> The key conceptual differences are:
> 1. Prior virtual nodes are learned / trainable constructs that require specifying their number as a key hyperparameter (e.g., K anchors or centroids). In contrast, NEUTAG’s feature nodes are not learned, are deterministically derived from the input feature matrix, and do not introduce additional parameters or architectural choices.
> 2. As shown in theorem 2 of menuscript, this feature-based construction provides a much tigher approximation bounds than GOAT whose bounds scale with **O(1/N)** vs NEUTAG's whose scales as follows.
> $$
> \mathbb{P}\left(\|\tilde{A}XW_V - AXW_V\|_F \leq \varepsilon \|A\|_F \|XW_V\|_F\right) > 1 - O(1/\exp(F))
> $$ Here, $F$ is the number of node features. Typically $exp(F)$ is **much higher than the number of nodes**, making the error bound **exponentially unlikely to fail** even in large graphs. This results in a **more reliable and tighter guarantee** across both small and massive graphs, which is not offered by prior virtual node based methods.
>
> >**The experimental results do not clearly demonstrate the effectiveness of the proposed Graph Transformer method compared to existing GT approaches.**
>
>
> **Response**
> We respectfully disagree with the concern that the empirical results are unclear.
> As highlighted in the table 1 of manuscript: NEUTAG ranks either 1st or 2nd on all datasets,  across $7$ datasets, performs consistently across both homophilic and heterophilc domains. Moreover, NEUTAG is the only sparse graph transformers that scale to dataset like snap-patents and preserve performance to small scale datasets too.
> We also had conducted additional experiments on 5 more challenging heterophilic datasets to show NEUTAG's effectiveness in table 7 of appendix. We request the reviewer to refer to appendix for the same. In these $5$ additional datasets, NEUTAG ranks 1.
> >**Some recent Graph Transformer methods [1–5] that also utilize virtual nodes are not discussed.**
>
> **Response**  We thank the reviewer for referring to these works. These methods explore global virtual nodes either using clustering or anchor sets. While we are unable to add direct comparision with these papers as their code-bases are not open-source, our comparision with GOAT (ICML 2023) and LargeGT which are also based on similar concept of clustering based virtual nodes, should be sufficient to show NEUTAG's effectivess with respect to these baselines. We have added these five works in the related work section of menuscript highlighted in blue.
>
>
> >**The paper requires careful proofreading and polishing.**
>
> **Response** We appreciate the reviewer’s suggestion and are happy to incorporate any specific issues the reviewer may point out. We have already revised the manuscript for clarity, grammar, and flow.
>
> ## **Summary**
>
> NEUTAG is, to the best of our knowledge, the first Graph Transformer to use feature-induced, deterministic, non-learned virtual nodes, enabling
>
> 1. A tighter theoretical approximation to full attention,
> 2. Assumption-free scalability to million-node graphs without sacrificing performance, and
> 3. Consistent performance across homophilic, heterophilic, small, and large graphs.
>
> We hope this clarifies the novelty and impact of our contribution and addresses the reviewer’s concerns.

---

### Note · Authors · 2026-01-02

**Comment:**

We are withdrawing our submission from ICLR. We sincerely thank the reviewers for their detailed reviews and valuable comments, and we appreciate the time and effort invested by the reviewers and organizers.

**Withdrawal Confirmation:**

I have read and agree with the venue's withdrawal policy on behalf of myself and my co-authors.